# POME: Post Optimization Model Edit via Muon-style Projection

## Abstract

We introduce Post-Optimization Model Edit (POME), a new algorithm that enhances the performance of fine-tuned large language models using only their pretrained and fine-tuned checkpoints, without requiring extra data or further optimization. The core idea is to apply a muon-style projection to $\Delta W$, the difference between the fine-tuned and pretrained weights. This projection uses truncated singular value decomposition (SVD) to equalize the influence of dominant update directions and prune small singular values, which often represent noise. As a simple post-processing step, POME is completely decoupled from the training pipeline. It requires zero modifications and imposes no overhead, making it universally compatible with any optimizer or distributed framework. POME delivers consistent gains, boosting average performance by +2.5% on GSM8K and +1.0% on code generation. Its broad applicability—from 7B foundation models to 72B RLHF-instructed models—establishes it as a practical, zero-cost enhancement for any fine-tuning pipeline.

## 1 Introduction

The pretrain–finetune paradigm has been widely adopted: given pretrained model weights, a model can be adapted to domain-specific datasets (e.g., math or coding) via supervised fine-tuning (SFT) or reinforcement learning (RL) to enhance specialized abilities (Schulman et al., 2017; Ouyang et al., 2022). Let $W_{\text{pre}}$ denote the pretrained weights and $W_{\text{ft}}$ the weights after fine-tuning, we pose the following question:

*Can we further improve the model using only $W_{pre}$ and $W_{ft}$, without any additional data or training?*

At first glance, this seems impossible since no new information is introduced. Surprisingly, we show that it is achievable through a novel algorithm we call **P**ost **O**ptimization **M**odel **E**dit (**POME**).

The key insight comes from recent advances in matrix-based optimizers, which often apply transformations to gradients or momentum to produce better updates (Jordan et al., 2024; Liu et al., 2025a; Team et al., 2025). For example, the Muon optimizer (Jordan et al., 2024) performs orthogonal projections on momentum at each step, ensuring that update directions contribute more uniformly rather than being dominated by a few principal components—a method validated in several recent LLM training works. Intuitively, we can treat $\Delta W = W_{\text{ft}} - W_{\text{pre}}$ as an aggregated update from the base optimizer (e.g., Adam across many steps), and it becomes plausible to refine this direction through a similar matrix-based transformation.

We instantiate this idea by applying truncated orthogonal projection, as in the Muon optimizer, to post-training weight edits. The intuition is that the benefits of orthogonalization in Muon do not fundamentally require per-step enforcement. The essential goal—ensuring that parameter changes span the space more uniformly—can also be achieved by projecting the final accumulated update. Therefore, given $\Delta W$, our method computes its orthogonal basis and quantizes its singular values to two levels (constant and 0). This retains the most significant update directions while suppressing noise from extremely high or low singular values. Crucially, this simple post-processing step requires no changes to training pipelines and remains compatible with any optimizer or distributed framework.

Empirical results demonstrate that POME consistently improves over standard fine-tuning across mathematical reasoning, code generation, and commonsense benchmarks, with ablations highlighting both its robustness and low computational cost. Specifically, POME achieves +2.5% on GSM8K and +0.7% on MATH for LLaMA2-7B (Touvron et al., 2023), +1.0% average improvement on HumanEval/MBPP code generation suites (Liu et al., 2024), and consistent gains across model scales from 7B to 72B parameters. The method also generalizes to RLHF-optimized models, improving Qwen2.5-Math (Yang et al., 2024) by +0.7% and Dr.GRPO (Liu et al., 2025c) by +2.1% on mathematical benchmarks.

Building on the theoretical insights and empirical findings, the main contributions of this paper are:

- We formalize post-training orthogonalization as a practical alternative to per-step matrix-aware training, showing that the key benefits can be achieved through one-time geometric correction of accumulated weight changes;
- We present POME, a training-free method that applies truncated SVD and spectrum equalization to weight deltas, requiring zero training-time overhead and no infrastructure modifications;
- We demonstrate consistent improvements over standard fine-tuning across mathematical reasoning, code generation, and commonsense benchmarks, with comprehensive ablations characterizing the method's robustness and computational cost.

## 2 PRELIMINARIES

### 2.1 MODEL EDIT

The enormous computational costs of large models make updating their knowledge far from straightforward. Ideally, as the world rapidly evolves in complex ways, a large model should keep pace in real time. Yet the heavy computing required to train a brand-new model makes instant updating infeasible. Consequently, a new paradigm "Model Edit" has emerged, enabling targeted modifications to a model's knowledge within specific domains while avoiding adverse effects on its behavior for other inputs. (Yao et al., 2023) Model-editing methods mainly fall into two groups: Modify internal parameters and Add extra parameters.

**Add extra parameters**: Keep the original weights fixed and add new parameters to handle the revised facts (Mitchell et al., 2022; Huang et al., 2023; Min, 2016; Li et al., 2025). **Modify internal parameters**: Update some weights with a $\Delta$ matrix. This can be done with "Locate-Then-Edit". Locate-Then-Edit first finds the key weights to change, then edits them (Sundararajan et al., 2017; Vig et al., 2020; Singh & Jaggi, 2020; Geva et al., 2020; Mitchell et al., 2021; Dai et al., 2021; Meng et al., 2022a;b; Tan et al., 2023; Hase et al., 2023; Jiang et al., 2025; Liu et al., 2025b). However, these methods primarily address knowledge editing across multiple tasks, focusing on updating large language models with knowledge derived from diverse task domains. In addition, model soup (Wortsman et al., 2022), ensemble learning (Dietterich et al., 2002), or EMA is another paradigm for post optimization model editing, but it relies on training multiple models.

### 2.2 MUON

Muon (Jordan et al., 2024) is an optimizer designed for the 2D parameters of the hidden layers in neural networks. It processes the updates generated by SGD with momentum by applying a Newton-Schulz iteration as a post-processing step to each update before applying them to the parameters.

Given a parameter $W \in \mathbb{R}^{m \times n}$, gradient $g \in \mathbb{R}^{m \times n}$, and momentum $M \in \mathbb{R}^{m \times n}$, the update rule is:

$$M_t = \mu M_{t-1} + g_t, \tag{1}$$

$$U_t = \text{NewtonSchulz}(M_t), \tag{2}$$

$$W_t = W_{t-1} - \eta(U_t + \lambda W_{t-1}). \tag{3}$$

Here, the Newton-Schulz function is an iterative algorithm that approximates the closest semi-orthogonal matrix to $M_t$, and the final update $U_t$ represents a modified version of $M_t$, where all singular values of $M_t$ are normalized to 1.

This approach ensures that, regardless of the original magnitude of the singular values, each independent direction receives equal scaling. This adjustment prevents the overlooking of critical directions during training, thereby enhancing the efficiency of parameter space exploration.

### 2.3 MATRIX ORTHOGONALIZATION

As previously discussed, Muon employs the Newton-Schulz iteration to compute an approximately semi-orthogonal matrix from $M_t$ by solving:

$$\text{Orthogonal}(M_t) = \arg\min_{U_t^\top U_t = I} \|M_t - U_t\|_F^2, \tag{4}$$

where $U_t$ is a semi-orthogonal matrix satisfying $U_t^\top U_t = I$, and $\|\cdot\|_F$ denotes the Frobenius norm.

Although Muon has demonstrated promising results (Jordan et al., 2024; Liu et al., 2025a; Team et al., 2025), its per-step matrix operations are impractical for large-scale distributed fine-tuning. In ZeRO/FSDP-style training (Rajbhandari et al., 2020; Paszke et al., 2019; Zhao et al., 2023), parameters are sharded across devices; even approximate orthogonalization (e.g., Newton–Schulz) must operate on full matrices, requiring frequent all-gathers to reassemble shards and subsequent redistributions. Moreover, the approximation introduces iteration and stabilization hyperparameters, adding numerical fragility and engineering overhead. In contrast, our method applies a Muon-like operator only once after post-training, avoiding the challenges associated with applying Muon at every step.

## 3 POME: POST-OPTIMIZATION MODEL EDIT

This section introduces our post-training orthogonalization approach. We first establish the problem setup and theoretical motivation in Section 3.1, then present our core method including SVD decomposition, rank truncation, and orthogonalization in Section 3.2. Finally, we analyze which layers benefit most from post-training orthogonalization through systematic layer-wise evaluation in Section 3.3.

### 3.1 MOTIVATION AND PROBLEM SETUP

We consider the post-training model edit setting. Formally, let $W_{\text{pre}}$ denote the parameters of the pretrained base model. After fine-tuning with a standard optimizer (e.g., Adam), we obtain

$$W_{\text{ft}} = W_{\text{pre}} + \Delta W, \quad \Delta W = W_{\text{ft}} - W_{\text{pre}}, \tag{5}$$

where $\Delta W = \{\Delta W_\ell\}_{\ell=1}^{L}$ collects the layer-wise weight updates. Our goal is to produce an edited model

$$W_E = W_{\text{pre}} + \text{Edit}(\Delta W), \tag{6}$$

where $\text{Edit}(\cdot)$ is a *data-free* post-hoc transformation of the fine-tuning update aimed at improving evaluation performance while controlling drift.

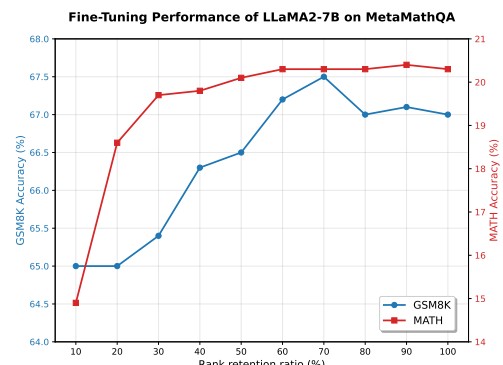

Figure 1: Effect of rank retention ratio on POME performance for LLaMA2-7B fine-tuned on MetaMathQA.

Our motivation builds on Muon (Sec. 2.2). Muon improves stability and generalization by equalizing the singular values of step wise updates, which shapes the update subspace during training. The same geometric effect can be applied once to the final accumulated update $\Delta W$. The central idea is to act directly on $\Delta W$ so that update energy is distributed more uniformly across principal directions, reducing interference from nearly collinear components without touching the training loop.

We next examine the spectrum of $\Delta W$. In this experiment, we take a finetuned model and truncate the trailing singular values while keep the leading singular values/vectors unchanged. The results in

Figure 1 indicate that the trailing singular vector spaces do not contribute much to the performance and the adaptation signal is concentrated in the leading subspace. This motivates a post training orthogonalization method that, for each selected layer, decomposes the update by singular value decomposition, keeps a compact subspace to suppress noise, and replaces the singular values of the retained components with a common level to rebalance directions. The full procedure is presented in Sec. 3.2.

## 3.2 OUR METHOD: POME

Inspired by the above empirical observations and the optimization objective from Muon (Yang et al., 2023; Bernstein, 2025), we formulate POME as a post-training edit that preserves RMS norm while enforcing low-rank structure. Specifically, the RMS-to-RMS operator norm measures the maximum factor by which a matrix amplifies the RMS norm of its input:

$$\|\Delta W\|_{\text{RMS}\to\text{RMS}} := \max_{x\neq 0} \frac{\|\Delta W x\|_{\text{RMS}}}{\|x\|_{\text{RMS}}} = \sqrt{\frac{\text{fan-in}}{\text{fan-out}}} \|\Delta W\|_*, \tag{7}$$

where $\|\cdot\|_*$ represents the spectral norm. Therefore, for the post-training edit, we define an optimization problem to stay close to the original weight matrix while ensuring the RMS-RMS norm is bounded by $\eta$ (Bernstein, 2025):

**Definition:** Let $\Delta W$ be the layer-wise weight matrix. We hope to find a matrix $P$, which can preserve RMS-RMS norm as Muon while being low-rank:

$$\max_{P:\text{rank}(P)\leq k} \langle \Delta W, P \rangle \quad \text{s.t. } \|P\|_{\text{RMS}\to\text{RMS}} \leq \eta,$$

where $\langle \cdot, \cdot \rangle$ represents the Frobenius inner product. The closed form solution can be written as

$$P^* = \eta\sqrt{\frac{\text{fan-out}}{\text{fan-in}}} U_k V_k^T,$$

where fan-out and fan-in represent the output and input dimensions of each layer, and $U_k V_k^T$ corresponds to setting the top-$k$ singular values to unity while zeroing out the rest. This is similar to the Muon formulation, except for the introduction of the low-rank constraint, which is important based on our empirical results.

The closed-form solution naturally suggests a practical algorithm: perform SVD decomposition on $\Delta W$, retain the top-$k$ singular vectors (as guided by Figure 1), and reconstruct with equalized singular values. This leads us to POME, a simple post-training procedure that implements this principle.

Concretely, for a target layer $\ell$ with delta $\Delta W_\ell \in \mathbb{R}^{m_\ell \times d_\ell}$ from standard fine-tuning, POME decomposes the update via SVD, retains the top-$k$ components based on our empirical analysis, and reconstructs with equalized singular values through the following procedure:

**Step 1: SVD decomposition (per selected layer).** For a target layer delta weight $\Delta W \in \mathbb{R}^{m \times d}$, compute its singular value decomposition:

$$\Delta W = U \Sigma V^\top, \tag{8}$$

where $U \in \mathbb{R}^{m \times m}$ and $V \in \mathbb{R}^{d \times d}$ are orthogonal, and $\Sigma = \text{diag}(\sigma_1 \geq \cdots \geq \sigma_r \geq 0)$.

**Step 2: Rank truncation.** Retain only the top-$k$ components to suppress noise/insignificant directions:

$$U_k = U[:, 1{:}k], \qquad V_k = V[:, 1{:}k], \tag{9}$$

**Step 3: Orthogonalization and Reconstruction.** Construct the orthogonalized delta by setting all retained singular values to unity and, optionally, apply a global scale $\alpha$:

$$\widehat{\Delta W}_\perp = U_k I_k V_k^\top = U_k V_k^\top, \quad \widehat{\Delta W} = \alpha \widehat{\Delta W}_\perp, \tag{10}$$

where $I_k$ is the $k \times k$ identity, $k$ can be set to $0.5 \cdot \text{rank}(\Delta W)$ since Figure 1 indicates that retaining the top 50% singular components recovers nearly all performance across most datasets, and $\alpha$ is

selected on a small held-out validation set to optimize performance. We provide the sensitivity analysis of $k$ and $\alpha$ in sec. A.2.

**Final edited model.** We keep unedited layers unchanged and replace selected layers by their orthogonalized deltas:

$$W_E = W_{\text{pre}} + \widehat{\Delta W}. \tag{11}$$

This simple post-processing procedure requires no modifications to the training process and can be applied to any fine-tuned model, making it highly practical for real-world deployment.

### 3.3 LAYER SELECTION

Given the data-free editing setup in Sec. 3.1 and our proposed edit method in Sec. 3.2, a practical question is *which layers to edit*. We therefore conduct a controlled, per–layer-type study to quantify sensitivity.

We evaluate on LLaMA2-7B, LLaMA3-8B and Gemma2-9B fine-tuned on MetaMathQA dataset. Only the target layer type is edited, and all others remain unchanged. We report accuracy deltas relative to the fine-tuned baseline.

| Model / Task | $Q_{\text{proj}}$ | $K_{\text{proj}}$ | $V_{\text{proj}}$ | $O_{\text{proj}}$ | $Gate_{\text{proj}}$ | $Up_{\text{proj}}$ | $Down_{\text{proj}}$ | Baseline |
|---|---|---|---|---|---|---|---|---|
| LLaMA2-7B/GSM8K | 67.1 | 68.2 | 67.9 | 68.1 | **68.9** | **69.7** | 68.2 | 67.2 |
| LLaMA2-7B/MATH | 19.7 | 19.3 | 19.4 | 20.0 | **19.8** | **19.7** | 19.7 | 19.4 |
| Average | 43.4 | 43.8 | 43.7 | 44.1 | **44.4** | **44.7** | 44.0 | 43.3 |
| LLaMA3-8B/GSM8K | 80.3 | 81.3 | 80.4 | 80.8 | **81.0** | **81.4** | 80.1 | 80.3 |
| LLaMA3-8B/MATH | 32.3 | 31.5 | 31.5 | 32.0 | **31.9** | **32.7** | 31.6 | 31.5 |
| Average | 56.3 | 56.4 | 56.0 | 56.4 | **56.5** | **57.1** | 55.9 | 55.9 |
| Gemma2-9B/GSM8K | 81.7 | 82.4 | 82.7 | 82.7 | 82.7 | **83.3** | **82.8** | 82.2 |
| Gemma2-9B/MATH | 36.0 | 36.7 | 36.8 | 37.2 | 37.3 | **37.3** | **38.9** | 36.1 |
| Average | 58.9 | 59.6 | 59.8 | 60.0 | 60.0 | **60.3** | **60.9** | 59.2 |

Table 1: Layer-wise post-training orthogonalization on LLaMA2-7B (MetaMathQA, lr=1e-5, seed=87, $k = 0.5 \cdot \text{rank}(\Delta W)$), LLaMA3-8B (MetaMathQA, lr=5e-6, seed=87, $k = 0.5 \cdot \text{rank}(\Delta W)$) and Gemma2-9B (MetaMathQA, lr=5e-6, seed=87, $k = 0.5 \cdot \text{rank}(\Delta W)$). Each column edits only the specified layer type and others remain unchanged.

Table 1 reports the layer-wise interventions. Two themes emerge consistently across tasks. First, attention projections ($Q_{\text{proj}}, K_{\text{proj}}, V_{\text{proj}}, O_{\text{proj}}$) are largely insensitive to post-hoc orthogonalization compared with FFN layers, suggesting their deltas are already well-conditioned. Second, the *FFN expansion* maps ($Up_{\text{proj}}$, $Down_{\text{proj}}$ and $Gate_{\text{proj}}$) benefit the most, aligning with our observation that high-dimensional expansions concentrate update energy into a few directions that orthogonalization can rebalance. Guided by these findings, our main experiments default to editing FFN expansion layers—using $Up_{\text{proj}}$ as the primary target.

## 4 EXPERIMENTS

We conduct comprehensive evaluations of POME across multiple experimental settings. We first evaluate supervised fine-tuning on mathematical reasoning, code generation, and commonsense reasoning tasks to establish baseline effectiveness. We then demonstrate the method's broader applicability by applying POME to publicly available fine-tuned models and models optimized with reinforcement learning from human feedback (RLHF). Finally, we provide detailed ablation studies to understand the contribution of key components.

---

**Algorithm 1** POME

---

**Require:** Base weights $W_{\text{pre}}$, delta $\Delta W$, scaling factor $\alpha$
**Ensure:** Edited model weights $W_E$
 1: **SVD decomposition**:
 2: $\qquad \Delta W = U \Sigma V^\top$
 3: **Rank truncation**:
 4: $\qquad U_k = U[:, 1{:}k], \quad V_k = V[:, 1{:}k],$
 5: **Orthogonalization and Reconstruction**:
 6: $\qquad \widehat{\Delta W}_\perp = U_k I_k V_k^\top = U_k V_k^\top, \quad \widehat{\Delta W} = \alpha \, \widehat{\Delta W}_\perp$
 7: **Final edited model**:
 8: $\qquad W_E = W_{\text{pre}} + \widehat{\Delta W}$
 9: **return** $W_E$

---

## 4.1 EXPERIMENTAL SETTING

**Mathematical Reasoning.** We fine-tune LLaMA2-7B (Touvron et al., 2023), LLaMA3-8B (Dubey et al., 2024), and Gemma2-9B models on MetaMathQA (Yu et al., 2023) and evaluate on GSM8K (Cobbe et al., 2021) and MATH (Hendrycks et al., 2021) benchmarks, which assess mathematical problem-solving capabilities.

**Code Generation.** We employ Code-Feedback (Zheng et al., 2024) as the training dataset for LLaMA2-7B and LLaMA3-8B, evaluating on HumanEval (Chen et al., 2021), HumanEval+, MBPP, and MBPP+ from EvalPlus (Liu et al., 2024).

**Commonsense Reasoning.** Following LLM-adapters (Hu et al., 2023), we train on Commonsense-170k and evaluate across eight benchmarks: BoolQ (Clark et al., 2019), PIQA (Bisk et al., 2020), SIQA (Sap et al., 2019), HellaSwag (Zellers et al., 2019), WinoGrande (Sakaguchi et al., 2021), ARC-C (Clark et al., 2018), ARC-E (Clark et al., 2018), and OBQA (Mihaylov et al., 2018).

**RLHF:** We investigate POME's applicability beyond supervised fine-tuning and evaluate on models optimized with reinforcement learning from human feedback (RLHF), such as open-sourced Qwen2.5-Math-72B-Instruct (Yang et al., 2024) model and variants of GRPO (Shao et al., 2024) (Dr.GRPO (Liu et al., 2025c)).

To ensure robustness, we conduct multiple runs with different random seeds and report averaged results where applicable.

## 4.2 MATHEMATICAL REASONING

We evaluate POME on mathematical reasoning tasks by fine-tuning three model families on Meta-MathQA. To ensure robustness, we train each model with three different random seeds and analyze performance across runs.

| Model | Task | $\text{Adam}_{42}$ | +POME | $\text{Adam}_{87}$ | +POME | $\text{Adam}_{100}$ | +POME |
|---|---|---|---|---|---|---|---|
| **LLaMA2-7B** | **GSM8K** | 66.9 | **69.0** (↑ **2.1**) | 67.2 | **69.7** (↑ **2.5**) | 66.9 | **69.9** (↑ **3.0**) |
| **LLaMA2-7B** | **MATH** | 19.0 | **19.8** (↑ **0.8**) | 19.4 | **19.7** (↑ **0.3**) | 19.0 | **20.0** (↑ **1.0**) |
| **LLaMA3-8B** | **GSM8K** | 80.6 | **80.9** (↑ **0.3**) | 80.3 | **81.4** (↑ **1.1**) | 81.3 | **81.7** (↑ **0.4**) |
| **LLaMA3-8B** | **MATH** | 31.8 | **32.7** (↑ **0.9**) | 31.5 | **32.7** (↑ **1.2**) | 31.6 | **32.9** (↑ **1.3**) |
| **Gemma2-9B** | **GSM8K** | 81.0 | **82.6** (↑ **1.6**) | 82.2 | **83.3** (↑ **1.1**) | 81.5 | **82.7** (↑ **1.2**) |
| **Gemma2-9B** | **MATH** | 36.4 | **36.9** (↑ **0.5**) | 36.1 | **37.3** (↑ **1.2**) | 37.3 | **38.3** (↑ **1.0**) |

Table 2: Accuracy of Fine-Tuning LLaMA2-7b, LLaMA3-8B and Gemma2-9B on MetaMathQA. $\text{Adam}_{42}$ represents the results with seed=42 and Adam optimizer.

Table 2 illustrates that POME achieves consistent improvements across models and seeds. For instance, on LLaMA2-7B with seed 87, POME improves GSM8K accuracy from 67.2% to 69.7%

and MATH accuracy from 19.4% to 19.7%. Similar gains are observed across LLaMA3-8B and Gemma2-9B, demonstrating the method's generalizability.

To further validate our approach beyond the models trained by ourselves, we apply POME to publicly available fine-tuned models. Table 3 demonstrates that POME improves MetaMath-Mistral-7B [1] performance from 77.9% to 79.5% on GSM8K and from 28.1% to 28.5% on MATH, confirming effectiveness on externally trained checkpoints.

| Model | Adam | +POME | GSM8K | MATH | Average |
|-------|------|-------|-------|------|---------|
| **MetaMath-Mistral-7B** | ✓ | | 77.9 | 28.1 | 53.0 |
| **MetaMath-Mistral-7B** | ✓ | ✓ | **79.5**(↑ **1.6**) | **28.5**(↑ **0.4**) | **54.0**(↑ **1.0**) |

Table 3: Effect of POME on the publicly available MetaMath-Mistral-7B model, demonstrating generalization to externally trained checkpoints.

## 4.3 CODE GENERATION

We assess POME's effectiveness on code synthesis tasks using two training datasets. Table 4 presents results for models trained on Code-Feedback (Zheng et al., 2024) and Magicoder-Evol-Instruct-110K (Wei et al., 2023) datasets.

On Code-Feedback, POME consistently improves performance across all evaluation metrics. For LLaMA2-7B, we observe gains of 0.6% on HumanEval, 1.2% on HumanEval+, 0.8% on MBPP, and 1.3% on MBPP+, yielding a 1.0% average improvement. On Magicoder-Evol-Instruct-110K, POME achieves more substantial gains, with a 1.7% average improvement across metrics.

| Model | Dataset | Method | HumanEval | HumanEval+ | MBPP | MBPP+ | Average |
|-------|---------|--------|-----------|------------|------|-------|---------|
| **LLaMA2-7B** | **CodeFeedback** | **Adam** | 34.8 | 32.3 | 40.2 | 32.8 | 35.0 |
| **LLaMA2-7B** | **CodeFeedback** | **+POME** | **35.4** (↑ **0.6**) | **33.5** (↑ **1.2**) | **41.0** (↑ **0.8**) | **34.1** (↑ **1.3**) | **36.0** (↑ **1.0**) |
| **LLaMA3-8B** | **CodeFeedback** | **Adam** | 57.3 | 52.4 | 64.6 | 56.1 | 57.6 |
| **LLaMA3-8B** | **CodeFeedback** | **+POME** | **58.6** (↑ **1.3**) | **53.6** (↑ **1.2**) | **65.2** (↑ **0.6**) | **58.3** (↑ **2.2**) | **58.9** (↑ **1.3**) |
| **LLaMA2-7B** | **Magicoder-Evol** | **Adam** | 47.0 | 43.3 | 31.5 | 27.2 | 37.3 |
| **LLaMA2-7B** | **Magicoder-Evol** | **+POME** | **49.4** (↑ **2.4**) | **43.3** (↑ **0.0**) | **33.9** (↑ **2.4**) | **29.4** (↑ **2.2**) | **39.0** (↑ **1.7**) |

Table 4: Accuracy of Fine-Tuning LLaMA2-7B and LLaMA3-8B (lr=5e-6, seed=87, $k = 0.5 \cdot \mathrm{rank}(\Delta W)$) on Code Generation Task.

## 4.4 COMMONSENSE REASONING

We evaluate POME on commonsense reasoning by fine-tuning LLaMA2-7B and LLaMA3-8B on Commonsense-170k and testing across eight downstream tasks. Table 5 presents comprehensive results across diverse reasoning benchmarks.

| Model | Method | BoolQ | PIQA | SIQA | HellaS | WinoG | ARC-C | ARC-E | OBQA | Average |
|-------|--------|-------|------|------|--------|-------|-------|-------|------|---------|
| **LLaMA2-7B** | **Adam** | **73.4** | 83.4 | **80.1** | 89.4 | 83.3 | 70.4 | 83.9 | 81.4 | 80.7 |
| **LLaMA2-7B** | **+POME** | 72.0 | **84.3** | **80.1** | **91.6** | **84.4** | **71.2** | **85.7** | **82.8** | **81.5** (↑ **0.8**) |
| **LLaMA3-8B** | **Adam** | 75.7 | 88.0 | 79.4 | 95.1 | 85.5 | 78.5 | 90.1 | 86.4 | 84.8 |
| **LLaMA3-8B** | **+POME** | **75.9** | **88.3** | **79.9** | **95.5** | **86.7** | 81.6 | **90.5** | **87.6** | **85.8** (↑ **1.0**) |

Table 5: Accuracy of Fine-Tuning LLaMA2-7b and LLaMA3-8B (lr=5e-6, seed=87, $k = 0.5 \cdot \mathrm{rank}(\Delta W)$) on Commonsense-170k.

---

[1]https://huggingface.co/meta-math/MetaMath-Mistral-7B

POME achieves consistent improvements across both model scales. For LLaMA2-7B, we observe notable improvements on HellaS (+2.2%) and ARC-E (+1.8%), yielding an overall average improvement of 0.8%. Similarly, LLaMA3-8B shows improvements on all eight benchmarks, with the largest gain on ARC-C (+3.1%) and an average improvement of 1.0%. The consistent improvements across diverse reasoning tasks demonstrate POME's effectiveness in enhancing response for commonsense inference.

## 4.5 RLHF Model

To investigate POME's applicability beyond supervised fine-tuning, we evaluate on models optimized with reinforcement learning from human feedback (RLHF). We apply POME to Qwen-2.5-Math models by computing weight deltas relative to their base checkpoints.

Table 6 demonstrates that POME improves performance on 6 out of 7 tasks for the 7B model and 5 out of 7 tasks for the 72B model, confirming effectiveness across model scales and optimization paradigms. For Qwen2.5-Math-7B-Instruct [2], POME increases average performance from 62.7% to 63.4%.

| Model | GSM8K | MATH | Minerva Math | Gaokao 2023 EN | Olympiad Bench | College Math | MMLU STEM | Average |
|---|---|---|---|---|---|---|---|---|
| Qwen2.5-Math-72B | **95.8** | 86.0 | 43.8 | 71.9 | 48.3 | **49.6** | 80.2 | 67.9 |
| Qwen2.5-Math-72B + POME | 95.5 | **86.7** | **44.5** | **73.8** | **50.4** | 49.3 | **80.4** | **68.7** (↑ **0.8**) |
| Qwen2.5-Math-7B | 95.8 | **83.5** | 33.8 | 67.8 | 41.5 | 46.9 | **69.3** | 62.7 |
| Qwen2.5-Math-7B + POME | **96.0** | **83.5** | **36.0** | **69.6** | **42.2** | **47.1** | 69.2 | **63.4** (↑ **0.7**) |

Table 6: Performance of Qwen2.5-Math-72B-Instruct models with and without POME across mathematical reasoning benchmarks.

Additionally, we evaluate POME on recent variants of GRPO (Shao et al., 2024), specifically Dr.GRPO [3] (Liu et al., 2025c) models. Table 7 shows that POME consistently improves performance across mathematical reasoning benchmarks. For Dr.GRPO, we observe notable improvements on AIME (43.3% → 46.7%) and AMC (59.0% → 66.3%), with gains also observed on Olympiad Bench. The average performance increases from 51.2% to 53.3%, demonstrating POME's effectiveness on advanced RL-optimized models.

| Model | AIME24 | AMC | MATH500 | Minerva | OlympiadBench | Average |
|---|---|---|---|---|---|---|
| **Dr.GRPO** | 43.3 | 59.0 | **80.2** | **31.6** | 41.9 | 51.2 |
| **Dr.GRPO + POME** | **46.7** | **66.3** | 79.6 | 31.3 | **42.5** | **53.3**(↑ **2.1**) |

Table 7: Accuracy comparison between baseline Dr.GRPO and Dr.GRPO with our proposed POME method on Qwen2.5-Math.

## 4.6 Generalization Analysis

To understand why POME improves model performance, we analyze its effect on generalization by examining both training accuracy and test accuracy. Table 8 presents results for LLaMA3-8B and Gemma2-9B trained on MetaMathQA datasets.

Our analysis reveals that POME consistently improves test accuracy *almost* without affecting training accuracy, suggesting that the method enhances generalization rather than simply overfitting to training data. For MetaMathQA, while training accuracy remains comparable between baseline and POME-edited models, test accuracy shows notable improvements. These findings suggest that

---

[2]https://huggingface.co/Qwen/Qwen2.5-Math-7B-Instruct
[3]https://huggingface.co/sail/Qwen2.5-Math-7B-Oat-Zero

| Model | Task | POME | Training Accuracy | Test Accuracy |
|-------|------|------|-------------------|---------------|
| LLaMA3-8B | GSM8K |  | 94.3 | 80.3 |
| LLaMA3-8B | GSM8K | ✓ | 94.2 (↓ 0.1) | 81.4 (↑ 1.1) |
| LLaMA3-8B | MATH |  | 54.7 | 31.5 |
| LLaMA3-8B | MATH | ✓ | 54.5 (↓ 0.2) | 32.7 (↑ 1.2) |
| Gemma2-9B | GSM8K |  | 95.8 | 82.2 |
| Gemma2-9B | GSM8K | ✓ | 95.9 (↑ 0.1) | 83.3 (↑ 1.1) |
| Gemma2-9B | MATH |  | 63.1 | 36.1 |
| Gemma2-9B | MATH | ✓ | 62.8 (↓ 0.3) | 37.3 (↑ 1.2) |

Table 8: Generalization analysis of POME on LLaMA3-8B and Gemma2-9B fine-tuned on Meta-MathQA (lr=5e-6, seed=87, $k = 0.5 \cdot \mathrm{rank}(\Delta W)$). Training accuracy and test accuracy demonstrate the method's effect on model generalization.

POME's effectiveness stems from improved parameter space utilization rather than memorization of training examples, providing insight into the mechanism underlying the observed performance gains.

## 4.7 ABLATION STUDY

**Rank Truncation and Equalization Analysis.** We examine the effect of retaining only top-$k$ singular components versus using all components (Truncation) and setting all retained singular values to unity (Equalization). Table 9 shows that while equalization yields improvements (e.g., 67.2% → 69.0% on GSM8K for LLaMA2-7B), rank truncation provides additional gains (67.2% → 69.7%), confirming that noise suppression contributes to performance.

| Model | Truncation | Equalization | GSM8K | MATH | Average |
|-------|-----------|--------------|-------|------|---------|
| LLaMA2-7B |  |  | 67.2 | 19.4 | 43.3 |
| LLaMA2-7B |  | ✓ | 69.0 | 19.6 | **44.3(↑ 1.0)** |
| LLaMA2-7B | ✓ | ✓ | 69.7 | 19.7 | **44.7(↑ 1.4)** |
| LLaMA3-8B |  |  | 80.3 | 31.5 | 55.9 |
| LLaMA3-8B |  | ✓ | 81.0 | 32.1 | **56.6(↑ 0.7)** |
| LLaMA3-8B | ✓ | ✓ | 81.4 | 32.7 | **57.1(↑ 1.2)** |

Table 9: Ablation study on the effects of truncation and POME on LLaMA2-7B (MetaMathQA, lr=1e-5, seed=87, $k = 0.5 \cdot \mathrm{rank}(\Delta W)$) and LLaMA3-8B (MetaMathQA, lr=5e-6, seed=87, $k = 0.5 \cdot \mathrm{rank}(\Delta W)$). The checkmarks indicate: Truncation (selecting top-50% principal components) and Equalization.

**Comparison with Baseline Methods.** Table 10 compares POME against training-time alternatives including NEFTune (Jain et al., 2023) and Exponential Moving Average (EMA). NEFTune achieves modest improvements across both models and tasks, while EMA shows mixed results with some performance degradation on certain benchmarks. POME consistently outperforms both methods, achieving the largest improvements across all settings. The limited effectiveness of EMA can be attributed to the similarity of models from the same training trajectory, providing minimal additional information for ensemble averaging. NEFTune's modest gains suggest that noise injection during training provides some regularization benefits, but lacks the targeted geometric correction that POME provides. In contrast, POME's superior performance stems from its ability to reshape the learned parameter subspace post-training, addressing directional imbalances that accumulate during optimization without requiring training-time modifications or hyperparameter tuning.

| Model | Task | Adam | NEFTune | EMA | POME |
|-------|------|------|---------|-----|------|
| **LLaMA2-7B** | **GSM8K** | 67.2 | 67.7 | 67.5 | **69.7** |
| **LLaMA2-7B** | **MATH** | 19.4 | **19.7** | 18.8 | **19.7** |
| **LLaMA3-8B** | **GSM8K** | 80.3 | 80.9 | 80.3 | **81.4** |
| **LLaMA3-8B** | **MATH** | 31.5 | 31.8 | 31.5 | **32.7** |

Table 10: Comparison between POME and baseline methods on LLaMA2-7B and LLaMA3-8B.

## 5 CONCLUSION

In this paper, we introduce POME, a training-free post-optimization method that applies matrix orthogonalization to fine-tuned language models through truncated SVD and spectrum equalization of weight deltas. By shifting orthogonalization from per-step training operations to a one-shot post-processing approach, POME eliminates the scalability issues of methods like Muon while preserving their geometric benefits. Experiments across mathematical reasoning, code generation, and commonsense tasks demonstrate consistent improvements over standard fine-tuning with zero training overhead. Our ablation studies confirm the method's robustness and identify FFN expansion layers as optimal targets for orthogonalization. POME's simplicity and broad applicability establish post-hoc subspace shaping as a practical alternative to matrix-aware training methods.

## 6 USAGE OF LARGE LANGUAGE MODELS

We declare that all core contributions of this work, including the research ideas, methodology design, experimental setup, and result analysis, were conceived and executed entirely by the authors through our own thinking and discussions. Large language models were used solely for language polishing and improving the clarity of our written presentation. Specifically, we used LLMs to refine sentence structure, grammar, and word choice in our manuscript, but no LLM-generated content was used for the technical substance, experimental design, or scientific insights presented in this paper.

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

# A APPENDIX

## A.1 IMPLEMENTATION DETAILS

Training is conducted on Nvidia H100 and H200 GPUs using BFloat16 precision. We set weight decay to 0 and employ a cosine learning rate scheduler with a 0.03 ratio linear warmup. For evaluation, we utilize vLLM (Kwon et al., 2023) to conduct our tests, ensuring efficient and scalable inference.

| Model | Dataset | LR | LR Scheduler | Warmup | Epochs | BS | Layer | k |
|---|---|---|---|---|---|---|---|---|
| **LLaMA2-7B** | MetaMathQA | 1e-5 | cosine | 300 | 3 | 128 | UP$_{\text{proj}}$ | 0.5 |
| **LLaMA2-7B** | commonsense-170k | 1e-5 | cosine | 300 | 3 | 32 | UP$_{\text{proj}}$ | 0.5 |
| **LLaMA2-7B** | Code-Feedback | 1e-5 | cosine | 300 | 3 | 32 | UP$_{\text{proj}}$ | 0.5 |
| **LLaMA3-8B** | MetaMathQA | 5e-6 | cosine | 300 | 3 | 128 | UP$_{\text{proj}}$ | 0.5 |
| **LLaMA3-8B** | commonsense-170k | 5e-6 | cosine | 300 | 3 | 32 | UP$_{\text{proj}}$ | 0.5 |
| **LLaMA3-8B** | Code-Feedback | 5e-6 | cosine | 300 | 3 | 32 | UP$_{\text{proj}}$ | 0.5 |
| **Gemma2-9B** | MetaMathQA | 5e-6 | cosine | 300 | 3 | 128 | UP$_{\text{proj}}$ | 0.5 |

Table 11: The Implementation Details of fine-tuning on LLaMA2-7B, LLaMA3-8B and Gemma2-9B.

## A.2 THE SENSITIVITY ANALYSIS OF K AND $\alpha$

In this section, we provide the sensitivity analysis of k (top-k components) and scale factor $\alpha$.

**The selection of K:** In our experiments, we define $k = K \cdot \text{rank}(\Delta W)$.

| Model | Task | K = 0.25 | K = 0.5 | K = 0.75 | K = 1.0 | Baseline |
|---|---|---|---|---|---|---|
| **LLaMA2-7B** | GSM8K | 69.7 | 69.7 | 69.1 | 69.2 | 67.2 |
| **LLaMA2-7B** | MATH | 19.9 | 19.7 | 19.6 | 19.7 | 19.4 |
| **Gemma2-9B** | GSM8K | 83.2 | 83.3 | 83.5 | 83.9 | 82.2 |
| **Gemma2-9B** | MATH | 37.8 | 37.3 | 37.3 | 36.9 | 36.1 |

Table 12: The sensitivity analysis of $k$ on MetaMathQA.

From the results, we can observe that POME is not sensitive to the selection of $k$. For example, in table 12, We observe that POME achieves consistent improvements when $K$ ranges from 0.25 to 0.75.

**The selection of $\alpha$:**

In our experiments, we first normalize the weight matrix and then apply a scaling factor $\beta$ to the weights:

$$\alpha = \beta * \|\Delta W\| \frac{\widehat{\Delta W_\perp}}{\|\widehat{\Delta W_\perp}\|} \tag{12}$$

| Model | Task | $\beta = 1.0$ | $\beta = 1.3$ | $\beta = 1.5$ | $\beta = 1.8$ | $\beta = 2.0$ | $\beta = 2.3$ | $\beta = 2.5$ | $\beta = 3.0$ |
|---|---|---|---|---|---|---|---|---|---|
| **LLaMA2-7B** | GSM8K | / | / | / | / | 68.9 | 68.3 | 68.9 | 68.2 |
| **LLaMA2-7B** | MATH | / | / | / | / | 19.6 | 19.2 | 19.6 | 19.7 |
| **LLaMA3-8B** | GSM8K | / | 80.6 | 81.4 | 81.2 | 81.4 | 80.7 | / | / |
| **LLaMA3-8B** | MATH | / | / | 32.7 | 32.1 | 32.1 | 31.4 | / | / |
| **Gemma2-9B** | GSM8K | 83.0 | 83.4 | 82.9 | 82.3 | / | / | / | / |
| **Gemma2-9B** | MATH | 37.3 | 36.9 | 36.6 | 35.9 | / | / | / | / |

Table 13: The sensitivity analysis of $\alpha$ on MetaMathQA ($k = 0.5 \cdot \text{rank}(\Delta W)$).

## A.3 THE SELECTION OF $\alpha$

In this section, we investigate how to determine $\alpha$ (defined in Equation 12) using training data. As shown in Table 14, the optimal $\alpha$ consistently corresponds to strong training performance. Therefore, we can simplify the selection process by using training performance as a metric to determine $\alpha$.

| Model | Task | $\beta = 1.0$ | $\beta = 1.3$ | $\beta = 1.5$ | $\beta = 1.8$ | $\beta = 2.3$ | $\beta = 2.5$ | $\beta = 3.0$ |
|---|---|---|---|---|---|---|---|---|
| **LLaMA2-7B** | GSM8K | / | / | / | / | 93.5/68.3 | 93.6/68.9 | 93.5/68.2 |
| **LLaMA2-7B** | MATH | / | / | / | / | 53.0/19.2 | 53.5/19.6 | 53.6/19.7 |
| **Gemma2-9B** | GSM8K | 95.5/83.0 | 95.9/83.4 | 95.9/82.9 | 95.8/82.3 | / | / | / |
| **Gemma2-9B** | MATH | 62.1/37.3 | 63.1/36.9 | 62.9/36.6 | 62.8/35.9 | / | / | / |

Table 14: The analysis (Train/Test) of $\alpha$ selection on LLaMA2-7B and Gemma2-9B.

## A.4 THEORETICAL ANALYSIS

In this section, our theoretical analysis is based on the theory in Muon (Bernstein, 2025).

**Problem.** Given a layer weight delta $\Delta W \in \mathbb{R}^{d_{\text{out}} \times d_{\text{in}}}$, we seek a low-rank linear operator $P$ that preserves the Muon RMS→RMS bound while aligning with $\Delta W$:

$$\max_{P:\text{rank}(P) \leq k} \langle \Delta W, P \rangle \quad \text{s.t. } \|P\|_{\text{RMS} \to \text{RMS}} \leq \eta. \tag{13}$$

(Here $d_{\text{in}}$ = fan-in, $d_{\text{out}}$ = fan-out.)

**Lemma 1** (RMS→RMS norm). *For any $A \in \mathbb{R}^{d_{out} \times d_{in}}$,*

$$\|P\|_{\text{RMS} \to \text{RMS}} = \sqrt{\frac{d_{in}}{d_{out}}} \ \|P\|_* \,,$$

*where $\|\cdot\|_*$ is the spectral norm.*

*Proof.* By definition, we can obtain:

$$\|P\|_{\text{RMS}} = \frac{1}{d} \|P\|_2 \,. \tag{14}$$

Then,

$$\|P\|_{\text{RMS} \to \text{RMS}} = \max_{x \neq 0} \frac{\|Px\|_{\text{RMS}}}{\|x\|_{\text{RMS}}} = \max_{x \neq 0} \frac{\|Px\|_2 / \sqrt{d_{\text{out}}}}{\|x\|_2 / \sqrt{d_{\text{in}}}} = \sqrt{\tfrac{d_{\text{in}}}{d_{\text{out}}}} \ \max_{x \neq 0} \frac{\|Px\|_2}{\|x\|_2} = \sqrt{\tfrac{d_{\text{in}}}{d_{\text{out}}}} \ \|P\|_* \,, \tag{15}$$

where $\|\cdot\|_*$ represents the spectral norm. By Lemma 1, letting

$$\tau := \eta\sqrt{\frac{d_{\text{out}}}{d_{\text{in}}}},$$

Problem (13) is equivalent to

$$\max_{P:\text{rank}(P)\leq k} \langle \Delta W, P \rangle \quad \text{s.t. } \|P\|_* \leq \tau. \tag{16}$$

Finally, we can transform the problem in equation 13 to equation 16 and solve the problem about equation 16 in Theorem 1.

**Theorem 1.** *Let the SVD of $\Delta W$ be $\Delta W = U\Sigma V^\top$, with singular values $\sigma_1 \geq \cdots$ and singular vectors $\{u_i, v_i\}$. Consider the problem*

$$\max_{P} \langle \Delta W, P \rangle \quad s.t. \ \|P\|_* \leq \tau, \ \text{rank}(P) \leq k. \tag{17}$$

*Let $U_k = [u_1, \ldots, u_k]$ and $V_k = [v_1, \ldots, v_k]$. Then an optimizer is*

$$P^\star = \tau \sum_{i=1}^{k} u_i v_i^\top = \eta\sqrt{\frac{d_{\text{out}}}{d_{\text{in}}}}\, U_k V_k^\top, \tag{18}$$

*and the optimal value equals*

$$\max_{\|P\|_2 \leq \tau, \ \text{rank}(P)\leq k} \langle \Delta W, P \rangle = \tau \sum_{i=1}^{k} \sigma_i(\Delta W). \tag{19}$$

*If $\sigma_k > \sigma_{k+1}$, the solution is unique up to sign flips of $\{u_i, v_i\}_{i \leq k}$; if there are ties at the $k$-th singular value, any orthonormal basis of the top-$k$ singular subspace is optimal.*

*Proof.* The feasible set $\{P : \|P\|_* \leq \tau, \ \text{rank}(P) \leq k\}$ is compact and nonempty, and the objective is linear, so a maximizer exists. Write the SVD $\Delta W = U\Sigma V^\top$ and set $Q := U^\top P V$. By unitary invariance of the inner product,

$$\langle \Delta W, P \rangle = \text{tr}(\Delta W^\top P) = \text{tr}(\Sigma Q).$$

By von Neumann's trace inequality (equivalently, Ky Fan's variational principle),

$$\text{tr}(\Sigma Q) \leq \sum_{i \geq 1} \sigma_i(\Delta W)\, \sigma_i(Q).$$

Since $\text{rank}(Q) \leq \text{rank}(P) \leq k$, we have $\sigma_i(Q) = 0$ for $i > k$. Also $\|Q\|_* = \|P\|_* \leq \tau$, hence $\sigma_i(Q) \leq \tau$ for all $i$. Therefore

$$\langle \Delta W, P \rangle = \sum_{i=1}^{k} \sigma_i(\Delta W)\, \sigma_i(Q) \leq \tau \sum_{i=1}^{k} \sigma_i(\Delta W). \tag{20}$$

The bound is tight when $Q$ aligns with the top-$k$ singular directions of $\Delta W$ and saturates the spectral-norm budget, i.e., $Q^\star = \tau \, \text{diag}(1, \ldots, 1, 0, \ldots)$ in the $(U, V)$ basis. Mapping back gives the maximizer

$$P^\star = UQ^\star V^\top = \tau \sum_{i=1}^{k} u_i v_i^\top, \tag{21}$$

which attains the value in equation 19. If $\sigma_k > \sigma_{k+1}$, the top-$k$ singular subspace is unique, so $P^\star$ is unique up to sign flips of the pairs $(u_i, v_i)$; under ties, any orthonormal basis of the top-$k$ subspace yields an optimizer.

Finally, rewriting $\tau$ via Lemma 1 (namely $\tau = \eta\sqrt{d_{\text{out}}/d_{\text{in}}}$) gives the stated form equation 18 and finish the proof.

