# OpenReview forum: "POME: Post Optimization Model Edit via Matrix Orthogonalization"
_ICLR.cc/2026/Conference — Submitted to ICLR 2026_

### Official Review · Reviewer_pP1m · 2025-10-20

**Soundness:** 2
**Presentation:** 1
**Contribution:** 3
**Rating:** 4
**Confidence:** 4

**Summary:**

The paper addresses the challenge of improving a fine-tuned large language model (LLM) post-training, using only the pretrained (W_pre) and fine-tuned (W_ft) checkpoints, without additional data or further optimization.

**Strengths:**

- The paper demonstrates strong originality by reimagining matrix orthogonalization (typically a per-step operation in optimizers like Muon) as a one-shot, training-free post-processing edit on accumulated weight deltas.
- Good experiments but with small models
- It offers an enhancement that could integrate into any LLM fine-tuning workflow. Raises the question, if we need to apply this, maybe we are not training models correctly? Maybe we need more regularization for post-training so we dont need to also apply this step?

**Weaknesses:**

The paper does a poor job at formalizing and conveying its main objective.

I believe the research question they are trying to address is (correct me if I misinterpreted it please): ``Can you take an already fine-tuned large language model and make it perform better after training is complete, using only the pretrained checkpoint (W_pre) and the fine-tuned checkpoint (W_ft), without any extra data, additional training steps, or modifications to the original training pipeline?'' This should be conveyed more effectively.

Discuss relation to other research works such as:

'LoRA: Low-Rank Adaptation of Large Language Models'
'Asymmetry in Low-Rank Adapters of Foundation Models'
'Make LoRA Great Again: Boosting LoRA with Adaptive Singular Values and Mixture-of-Experts Optimization Alignment'
'LoRA Fine-Tuning Without GPUs: A CPU-Efficient Meta-Generation Framework for LLMs'

Lora is a concurrent line of research and different from this paper, but it would help contextualize the contribution.

**Questions:**

Can the authors come up with any generalization theoretical results similar to:
'Asymmetry in Low-Rank Adapters of Foundation Models' ?

Can this scale to larger models?

---

> ### Author Response · Authors · 2025-11-28
> **Response to Reviewer pP1m**
>
> Thanks for your constructive and inspiring feedback, we carefully address your concerns below.
>
> >W1: The paper does a poor job at formalizing and conveying its main objective. I believe the research question they are trying to address is (correct me if I misinterpreted it please): "Can you take an already fine-tuned large language model and make it perform better after training is complete, using only the pretrained checkpoint (W_pre) and the fine-tuned checkpoint (W_ft), without any extra data, additional training steps, or modifications to the original training pipeline?" This should be conveyed more effectively.
>
> We sincerely apologize for the lack of clarity in our presentation. Your understanding is exactly correct. Our research question is: "Can we improve an already fine-tuned large language model's performance post-training, using only the pretrained checkpoint (W_pre) and fine-tuned checkpoint (W_ft), without requiring additional data, training steps, or modifications to the training pipeline?"
>
> We have revised the introduction and related sections to better articulate this objective and provide clearer exposition of our method. The updated version has been submitted, and we hope it addresses your concern and helps readers better understand our work. We appreciate your feedback in helping us improve the paper's clarity.
>
> >W2: LoRA is a concurrent line of research and different from this paper, but it would help contextualize the contribution.
>
> Thank you for this constructive comment. We appreciate the opportunity to clarify the relationship between POME and LoRA, as they address different aspects of model optimization:
>
> **(1) Different Stages and Purposes:** LoRA is a parameter-efficient fine-tuning method that reduces training costs by learning low-rank adapters during training. In contrast, POME is a post-training technique applied after fine-tuning is complete to further enhance model performance. They operate at different stages of the model development pipeline.
>
> **(2) Different Rank Characteristics:** While LoRA imposes strict low-rank constraints (typically r=8-64) during training, POME is not fundamentally a low-rank method. First, our ablation studies show that even without singular value truncation (K=1.0), orthogonalization alone improves performance (see Table below). Second, when we do apply truncation, we typically retain half of the rank (k=0.5·rank(ΔW)), which maintains substantially more representational capacity than typical LoRA configurations.
>
> | Model | Task | K = 0.25 | K = 0.5 | K = 0.75 | K = 1.0 | Baseline |
> |-------|------|----------|---------|----------|---------|----------|
> | LLaMA2-7B | GSM8K | 69.7 | 69.7 | 69.1 | 69.2 | 67.2 |
> | LLaMA2-7B | MATH | 19.9 | 19.7 | 19.6 | 19.7 | 19.4 |
>
> **(3) Different Performance Baselines:** LoRA typically achieves slightly lower performance than full fine-tuning due to its rank constraints, though it offers substantial training efficiency benefits. POME, on the other hand, is designed to improve upon full fine-tuning performance, targeting a different optimization objective.
>
> **(4) Orthogonality and Complementarity:** Importantly, POME and LoRA are orthogonal approaches that can be combined. We demonstrate this by applying POME to LoRA-tuned models with ranks 16 and 32 on LLaMA2-7B and LLaMA3-8B using the MetaMathQA dataset.
>
> | Model | Algorithm | Rank | GSM8K | MATH |
> |-------|-----------|------|-------|------|
> | LLaMA2-7B | Adam | 16 | 66.7 | 16.7 |
> | LLaMA2-7B | +POME | 16 | **67.6** | **17.5** |
> | LLaMA3-8B | Adam | 16 | 81.4 | 30.7 |
> | LLaMA3-8B | +POME | 16 | **82.2** | **31.2** |
>
> | Model | Algorithm | Rank | GSM8K | MATH |
> |-------|-----------|------|-------|------|
> | LLaMA2-7B | Adam | 32 | 66.0 | 17.5 |
> | LLaMA2-7B | +POME | 32 | **66.7** | **18.2** |
> | LLaMA3-8B | Adam | 32 | 81.7 | 31.6 |
> | LLaMA3-8B | +POME | 32 | **82.5** | **32.3** |
>
> The results show that POME consistently improves LoRA performance, demonstrating that practitioners can benefit from both efficient training (via LoRA) and post-training enhancement (via POME). This positions POME as a broadly applicable technique that complements various fine-tuning paradigms, from parameter-efficient methods to full fine-tuning.

---

> ### Author Response · Authors · 2025-11-28
> **Response to Reviewer pP1m - Continue**
>
> >Q1: Can the authors come up with any generalization theoretical results similar to: 'Asymmetry in Low-Rank Adapters of Foundation Models'?
>
> Thank you for this insightful suggestion. We agree that establishing a generalization theory similar to the referenced work would significantly strengthen our understanding of POME. We will add more discussion about LoRA and your provided important work "Asymmetry in Low-Rank Adapters of Foundation Models" in our new revision. While a complete theoretical framework remains future work, we offer both theoretical intuition and empirical evidence supporting POME's generalization benefits.
>
> **(1) Theoretical Intuition:** We can interpret POME through the lens of existing optimization theory. The Muon optimizer applies orthogonal projections to momentum at each optimization step. In POME, we can view ΔW = W_ft - W_pre as an accumulated update from the base optimizer (e.g., Adam) across N training steps. When N=1, POME reduces to a Muon-like single-step update, suggesting that Muon represents a special case of our method applied at each step. This connection indicates that generalization analyses of Muon-style optimizers (which leverage Newton-Schulz iteration for orthogonalization) could provide a theoretical foundation for understanding POME's properties. The key insight is that orthogonalization restructures the weight update to better align with the geometry of the loss landscape, potentially leading to flatter minima and improved generalization—a hypothesis supported by our empirical findings below.
>
> **(2) Empirical Evidence for Generalization:** We provide empirical results examining both training and test accuracy to demonstrate POME's effect on generalization. The table below presents results for LLaMA3-8B and Gemma2-9B fine-tuned on MetaMathQA.
>
> | Model | Task | POME | Training Accuracy | Test Accuracy |
> |-------|------|------|-------------------|---------------|
> | LLaMA3-8B | GSM8K | | 94.3 | 80.3 |
> | LLaMA3-8B | GSM8K | ✓ | 94.2 (↓ 0.1) | **81.4 (↑ 1.1)** |
> | LLaMA3-8B | MATH | | 54.7 | 31.5 |
> | LLaMA3-8B | MATH | ✓ | 54.5 (↓ 0.2) | **32.7 (↑ 1.2)** |
> | Gemma2-9B | GSM8K | | 95.8 | 82.2 |
> | Gemma2-9B | GSM8K | ✓ | 95.9 (↑ 0.1) | **83.3 (↑ 1.1)** |
> | Gemma2-9B | MATH | | 63.1 | 36.1 |
> | Gemma2-9B | MATH | ✓ | 62.8 (↓ 0.3) | **37.3 (↑ 1.2)** |
>
>
> The results reveal a consistent pattern: POME maintains or slightly decreases training accuracy while substantially improving test accuracy. This behavior suggests that POME's orthogonalization mechanism enhances generalization rather than simply memorizing training data. The improved test performance with comparable or lower training accuracy indicates that the method helps models find solutions that transfer better to out-of-distribution examples, which is a key property of good generalization.

---

> ### Author Response · Authors · 2025-11-28
> **Response to Reviewer pP1m - Continue**
>
> >Q2: Can this scale to larger models?
>
> Thank you for this constructive comment. We agree that demonstrating scalability is crucial for establishing POME's practical utility. We conducted extensive experiments across models ranging from 7B to 72B parameters to validate POME's effectiveness at scale.
>
> **(1) 32B Scale with RL Training:** We fine-tuned Qwen-32B-base using DAPO (a reinforcement learning algorithm) and evaluated on AIME. The results show that POME substantially improves DAPO's performance, achieving a 3.8-point gain (from 51.0 to 54.8).
>
> | Model | Algorithm | AIME |
> |-------|-----------|------|
> | Qwen-32B-base | DAPO | 51.0 |
> | Qwen-32B-base | DAPO+POME | **54.8 ($\uparrow$ 3.8)** |
>
> **(2) 70B Scale with Full Fine-tuning:** We fine-tuned LLaMA3-70B and evaluated on GSM8K, MATH, and CollegeMath. POME maintains its effectiveness at this scale, achieving an average 1.0-point improvement across mathematical reasoning benchmarks.
>
> | Model | Algorithm | GSM8K | MATH | CollegeMath |
> |-------|-----------|-------|------|-------------|
> | DART-Math-Llama3-70B | Adam | 54.9 | 90.4 | 38.5 |
> | DART-Math-Llama3-70B | Adam+POME | **56.3** | **91.8** | **39.5** |
>
> **(3) 72B Scale with RL Training:** We evaluated POME on Qwen2.5-Math-72B trained with reinforcement learning across a comprehensive benchmark suite. The results demonstrate consistent improvements across most evaluation tasks, with notable gains on challenging benchmarks like Minerva Math and Gaokao.
>
> | Model | Algorithm | GSM8K | MATH | Minerva Math | Gaokao 2023 EN | Olympiad Bench | College Math | MMLU |
> |-------|-----------|-------|------|--------------|----------------|----------------|--------------|------|
> | Qwen2.5-Math-72B | Adam | 95.8 | 86.0 | 43.8 | 71.9 | 48.3 | 49.6 | 80.2 | 67.9 |
> | Qwen2.5-Math-72B | Adam+POME | 95.5 | **86.7** | **44.5** | **73.8** | **50.4** | 49.3 |** 80.4** | **68.7** |
>
> These results demonstrate that POME scales effectively to models exceeding 70B parameters while maintaining computational efficiency. The method's training-free nature means that computational cost scales only with a single forward pass for SVD computation, making it practical even for very large models. The consistent improvements across different model sizes, training methods (supervised fine-tuning and RL), and benchmark difficulties confirm POME's broad applicability and scalability.

---

### Official Review · Reviewer_GCvj · 2025-10-25

**Soundness:** 3
**Presentation:** 2
**Contribution:** 2
**Rating:** 4
**Confidence:** 4

**Summary:**

This paper introduces the POME (Post Optimization Model Edit) technique. Inspired by Muon, POME can serve as a post fine-tuning technique, which optimizes the delta weight updates from the fine-tuning stage. It proposes the key insight that `the benefits of orthogonalization in Muon do not fundamentally require per-step enforcement'. POME demonstrates consistent performance gains on different model sizes and different post-training stages (fine-tuning to RLHF).

**Strengths:**

1. This paper proposes and validates a great insight: Muon's benefits can be achieved without requiring per-step enhancement. This can resolve some distributed training issues introduced by the Muon optimizer, while also improving fine-tuning performance.
2. Thorough experiments validate POME's advantage over vanilla fine-tuning on different datasets and settings.

**Weaknesses:**

1. Although I recognize this paper's contribution, the main intuition and most of the method details are adopted from Muon, which limits this work's contribution.
2. No direct comparison (both theoretically and experimentally) between POME and Muon-trained models' performance, readers cannot fully understand the trade-off between training efficiency/flexibility and performance.
3. The improvements are marginal compared with Adam/NEFTune. Additionally, the performance appears to be highly sensitive to the chosen layer and the retention rank ratio, which limits this method's usability.
4. Flawed presentation: e.g., the two methods in Table 7 are both 'Dr. DRPO'.

**Questions:**

1. Could the authors theoretically and experimentally (efficiency, performance) compare POME with Muon?
2. Could the authors provide more ablation experiments on the applied layers and the sensitivity to hyperparameters like learning rate?

---

> ### Author Response · Authors · 2025-11-28
> **Response to Reviewer GCvj**
>
> Thanks for your insightful comments, we carefully address your concerns below.
>
> >W1: Although I recognize this paper's contribution, the main intuition and most of the method details are adopted from Muon, which limits this work's contribution.
>
> Thank you for this detailed comment. While we acknowledge that POME draws inspiration from the orthogonal projection concept in Muon, we respectfully highlight several fundamental differences in both methodology and research focus:
>
> **(1) Different Research Focus:** Muon applies orthogonal projections to the momentum during training to improve convergence dynamics, whereas POME applies orthogonalization directly to the final weight deltas after training is complete. This represents a fundamentally different approach: training-time optimization versus post-training geometric correction.
>
> **(2) Different Use Case and Motivation:** Muon focuses on achieving faster convergence during training through matrix-aware updates at each step. In contrast, POME is entirely training-free and focuses on improving already-trained models through one-time geometric correction. This makes POME complementary to any training method, including Muon itself (as shown in our experiments where POME+Muon outperforms Muon alone).
>
> **(3) Novel Contributions:** We believe POME makes several distinct contributions:
> - We formalize post-training orthogonalization as a practical alternative to per-step matrix-aware training, demonstrating that key benefits can be achieved through one-time geometric correction of accumulated weight changes;
> - We present a training-free method that applies truncated SVD and spectrum equalization to weight deltas, requiring zero training-time overhead and no infrastructure modifications;
> - We provide comprehensive empirical validation across diverse domains (mathematical reasoning, code generation, commonsense QA) with extensive ablations characterizing the method's robustness, scalability to 70B+ parameters, compatibility with various training methods (Adam, Muon, LoRA, RL), and computational efficiency.
>
> These contributions establish POME as a distinct and practical approach that democratizes access to matrix-aware optimization benefits without requiring specialized training infrastructure.
>
> >W2: No direct comparison (both theoretically and experimentally) between POME and Muon-trained models' performance, readers cannot fully understand the trade-off between training efficiency/flexibility and performance.
>
> Thank you for this constructive comment. We agree that a direct comparison between POME and Muon is important for understanding their respective benefits. We conducted experiments fine-tuning LLaMA2-7B and LLaMA3-8B with Muon and applying POME to both Adam and Muon-trained models.
>
> | Model | Algorithm | GSM8K | MATH |
> |-------|-----------|-------|------|
> | LLaMA2-7B | Adam | 67.2 | 19.4 |
> | LLaMA2-7B | Adam+POME | **69.7** | **19.7** |
> | LLaMA2-7B | Muon | 67.5 | 19.2 |
> | LLaMA2-7B | Muon+POME | **68.3** | **19.5** |
>
> **Table: Accuracy of LLaMA2-7B on MetaMathQA.**
>
> | Model | Algorithm | GSM8K | MATH |
> |-------|-----------|-------|------|
> | LLaMA3-8B | Adam | 82.2 | 36.1 |
> | LLaMA3-8B | Adam+POME | **83.4** | **36.9** |
> | LLaMA3-8B | Muon | 82.6 | 35.9 |
> | LLaMA3-8B | Muon+POME | **83.6** | **36.2** |
>
> **Table: Accuracy of LLaMA3-8B on MetaMathQA.**
>
> The results reveal several key insights: (1) Adam+POME achieves comparable or better performance than Muon alone, demonstrating that post-training orthogonalization can match the benefits of training-time matrix-aware optimization; (2) POME is complementary to Muon—combining them yields the best overall performance; (3) POME offers greater flexibility as it can be applied post-hoc to any training method without requiring specialized infrastructure or increased training costs. This positions POME as both a standalone alternative and a complementary enhancement to matrix-aware optimizers.

---

> ### Author Response · Authors · 2025-11-28
> **Response to Reviewer GCvj - Continue**
>
> >W3: The improvements are marginal compared with Adam/NEFTune. Additionally, the performance appears to be highly sensitive to the chosen layer and the retention rank ratio, which limits this method's usability.
>
> Thank you for this comment. We would like to address both concerns regarding improvement magnitude and hyperparameter sensitivity.
>
> **(1) Magnitude of Improvements:** In Table 10 of our submission, we compared POME with baseline methods (NEFTune, EMA). The detailed results below show that POME achieves average improvements of 1.0 points for LLaMA2-7B and 0.7 points for LLaMA3-8B on mathematical reasoning tasks.
>
> | Model | Task | Adam | NEFTune | EMA | POME |
> |-------|------|------|---------|-----|------|
> | LLaMA2-7B | GSM8K | 67.2 | 67.7 | 67.5 | **69.7** |
> | LLaMA2-7B | MATH | 19.4 | **19.7** | 18.8 | **19.7** |
> | LLaMA2-7B | Avg | 43.3 | 43.7 | 43.2 | **44.7 (↑ 1.0)** |
> | LLaMA3-8B | GSM8K | 80.3 | 80.9 | 80.3 | **81.4** |
> | LLaMA3-8B | MATH | 31.5 | 31.8 | 31.5 | **32.7** |
> | LLaMA3-8B | Avg | 55.9 | 56.4 | 55.9 | **57.1 (↑ 0.7)** |
>
> To further verify POME's effectiveness, we provide results on code generation tasks. POME achieves gains of 0.7 points for LLaMA2-7B and 0.8 points for LLaMA3-8B, demonstrating consistent benefits across different domains.
>
> | Model | Task | Adam | NEFTune | EMA | POME |
> |-------|------|------|---------|-----|------|
> | LLaMA2-7B | HumanEval | 34.8 | 34.8 | 34.1 | **35.4** |
> | LLaMA2-7B | MBPP | 40.2 | 40.1 | 40.3 | **41.0** |
> | LLaMA2-7B | Avg | 37.5 | 37.5 | 37.2 | **38.2 (↑ 0.7)** |
> | LLaMA3-8B | HumanEval | 57.3 | 57.9 | 57.3 | **58.6** |
> | LLaMA3-8B | MBPP | 64.6 | 64.3 | 64.6 | **65.2** |
> | LLaMA3-8B | Avg | 61.0 | 61.1 | 61.0 | **61.9 (↑ 0.8)** |
>
> Importantly, POME is training-free and imposes zero computational overhead during training, making these consistent improvements practically valuable. Moreover, POME can be combined with training-time techniques like NEFTune or Muon for complementary benefits.
>
> **(2) Hyperparameter Sensitivity:** We conducted comprehensive sensitivity analyses for both k and α to demonstrate POME's robustness.
>
> **Truncation parameter k:** The results show that POME maintains consistent performance across different truncation ratios. Notably, even without truncation (K=1.0), POME improves baseline performance, with truncation providing additional gains. Performance remains stable when K ranges from 0.25 to 0.75.
>
> | Model | Task | K = 0.25 | K = 0.5 | K = 0.75 | K = 1.0 | Baseline |
> |-------|------|----------|---------|----------|---------|----------|
> | LLaMA2-7B | GSM8K | 69.7 | 69.7 | 69.1 | 69.2 | 67.2 |
> | LLaMA2-7B | MATH | 19.9 | 19.7 | 19.6 | 19.7 | 19.4 |
> | Gemma2-9B | GSM8K | 83.2 | 83.3 | 83.5 | 83.9 | 82.2 |
> | Gemma2-9B | MATH | 37.8 | 37.3 | 37.3 | 36.9 | 36.1 |
>
> **Scale factor $\alpha$:** We define $\alpha = \beta * \|\Delta W\| \frac{\Delta W_{\perp}}{\|\Delta W_{\perp}\|}.$and analyze sensitivity through the global parameter β. The results demonstrate robust performance across reasonable ranges. For example, LLaMA2-7B maintains consistent improvements when β ranges from 2.0 to 3.0.
>
> | Model | Task |  $\beta$  = 2.0 | $\beta$  = 2.3 | $\beta$  = 2.5 | $\beta$  = 3.0 |
> |-------|------|---------|---------|---------|---------|
> | LLaMA2-7B | GSM8K |  68.9 | 68.3 | 68.9 | 68.2 |
> | LLaMA2-7B | MATH |  19.6 | 19.2 | 19.6 | 19.7 |
>
>
>
> | Model | Task | $\beta$  = 1.0 | $\beta$  = 1.3 | $\beta$  = 1.5 | $\beta$  = 1.8 |
> |-------|------|---------|---------|---------|---------|
> | Gemma2-9B | GSM8K | 83.0 | 83.4 | 82.9 | 82.3 | / | / | / | / |
> | Gemma2-9B | MATH | 37.3 | 36.9 | 36.6 | 35.9 | / | / | / | / |
>
> >W4: Flawed presentation: e.g., the two methods in Table 7 are both 'Dr. DRPO'.
>
> We sincerely apologize for this error in Table 7. We have corrected this in the revised manuscript.
>
> To provide additional evidence of POME's effectiveness with reinforcement learning algorithms, we conducted experiments on AIME using Qwen2.5-32B with the DAPO algorithm. The results demonstrate that POME substantially improves DAPO's performance, achieving a 3.8-point gain (from 51.0 to 54.8).
>
> | Model | Algorithm | AIME |
> |-------|-----------|------|
> | Qwen2.5-32B | DAPO | 51.0 |
> | Qwen2.5-32B | DAPO+POME | **54.8 ($\uparrow$ 3.8)** |

---

> ### Author Response · Authors · 2025-11-28
> **Response to Reviewer GCvj - Continue**
>
> >Q1: Could the authors theoretically and experimentally (efficiency, performance) compare POME with Muon?
>
> Thank you for this constructive comment. We would like to clarify that POME and Muon are orthogonal methods applied at different stages of the training pipeline. Muon is an optimizer used during training, while POME is a training-free post-processing technique applied after obtaining the fine-tuned checkpoint. Although both leverage orthogonalization, they serve different purposes and can be combined for complementary benefits. Below we provide both performance and efficiency comparisons.
>
> **(1) Performance Comparison:** We fine-tuned LLaMA2-7B and LLaMA3-8B using Adam, Muon, and their combinations with POME on MetaMathQA. The results are shown below:
>
> | Model | Algorithm | GSM8K | MATH |
> |-------|-----------|-------|------|
> | LLaMA2-7B | Adam | 67.2 | 19.4 |
> | LLaMA2-7B | Muon | 67.5 | 19.2 |
> | LLaMA2-7B | Adam+POME | **69.7** | **19.7** |
> | LLaMA2-7B | Muon+POME | **68.3** | **19.5** |
>
>
> | Model | Algorithm | GSM8K | MATH |
> |-------|-----------|-------|------|
> | LLaMA3-8B | Adam | 82.2 | 36.1 |
> | LLaMA3-8B | Muon | 82.6 | 35.9 |
> | LLaMA3-8B | Adam+POME | **83.4** | **36.9** |
> | LLaMA3-8B | Muon+POME | **83.6** | **36.2** |
>
> Key observations:
> 1. POME consistently improves performance when applied to both Adam and Muon-trained models, demonstrating its complementary nature.
> 2. Adam and Muon achieve similar final performance, consistent with findings in related work [1,2,3] that Muon's primary advantage lies in convergence speed rather than final accuracy in some scenarios, particularly in post-training scenarios.
>
> **(2) Efficiency Comparison:** POME is training-free and introduces no training-time computational overhead. In contrast, Muon modifies the training process itself. We observed that Muon achieves faster convergence than Adam when fine-tuning LLaMA2-7B on MetaMathQA. Importantly, POME can be combined with Muon to obtain both fast convergence during training (from Muon) and improved final performance (from POME), offering the best of both approaches.
>
> [1] On the convergence analysis of muon.
>
> [2] On the Convergence of Muon and Beyond.
>
> [3] MUON IS SCALABLE FOR LLM TRAINING.

---

> ### Author Response · Authors · 2025-11-28
> **Response to Reviewer GCvj - Continue**
>
> >Q2: Could the authors provide more ablation experiments on the applied layers and the sensitivity to hyperparameters like learning rate?
>
> Thank you very much for your constructive comment.
>
> 1. More ablation experiments on the applied layers: We conduct experiments on code generation task to provide more ablation experiments on the applied layers. From the results in the table, we can notice that only edit FFN layers (up-projection) can also obtain the highest performance on HumanEval and MBPP.
>
> | Model / Task | Embed | Q_proj | K_proj | V_proj | O_proj | Gate_proj | Up_proj | Down_proj | Baseline |
> |--------------|-------|--------|--------|--------|--------|-----------|---------|-----------|----------|
> | LLaMA2-7B/HumanEval | 35.4 | 32.9 | 33.5 | 33.5 | 34.8 | 35.4 | 35.4 | 34.1 | 34.8 |
> | LLaMA2-7B/HumanEval+ | 32.3 | 31.1 | 31.7 | 32.3 | 31.7 | 33.5 | 33.5 | 31.7 | 32.3 |
> | LLaMA2-7B/MBPP | 40.5 | 41.0 | 41.3 | 41.0 | 40.5 | 40.7 | 41.0 | 40.5 | 40.2 |
> | LLaMA2-7B/MBPP+ | 32.8 | 33.6 | 33.6 | 34.1 | 34.1 | 34.1 | 34.1 | 33.6 | 32.8 |
> | Average | 35.3 | 34.7 | 35.0 | 35.2 | 35.3 | **35.9** | **36.0** | 35.0 | 35.0 |
>
> 2. Sensitivity to hyperparameters: To address the concern about the sensitive hyperparameters, we provide the sensitivity analysis of learning rate, k=K*Rank(ΔW) and $\alpha$ in the table.
>
> (1) To address the concern about the sensitivity of the learning rate selection. We conduct the experiments on applying POME into the models with different learning rates and analyze whether the performance gain from POME is robust to different learning rate selection. The results are shown in the table, we can obtain that POME can improve the performance of all these 3 models with different learning rates.
>
> | Model | Algorithm | Learning Rate | GSM8K | MATH |
> |-------|-----------|---------------|-------|------|
> | LLaMA2-7B | Adam | 1e-5 | 67.2 | 19.4 |
> | LLaMA2-7B | Adam+POME | 1e-5 | **69.7 (↑ 2.5)** | **19.7 (↑ 0.3)** |
> | LLaMA2-7B | Adam | 5e-6 | 66.0 | 18.2 |
> | LLaMA2-7B | Adam+POME | 5e-6 | **68.1 (↑ 2.1)** | **19.8 (↑ 1.6)** |
> | LLaMA2-7B | Adam | 2e-5 | 67.0 | 18.3 |
> | LLaMA2-7B | Adam+POME | 2e-5 | **68.6 (↑ 1.6)** | **19.6 (↑ 1.3)** |
>
> (2) From the results in the following table, we can observe that POME is not sensitive to the selection of k. For example, in the table below, we observe that POME achieves consistent improvements when K ranges from 0.25 to 0.75. In addition, we would like to claim that even if we do not use truncation (K=1.0), POME can also improve the vanilla model's performance, and using truncation can further improve its performance.
>
> | Model | Task | K = 0.25 | K = 0.5 | K = 0.75 | K = 1.0 | Baseline |
> |-------|------|----------|---------|----------|---------|----------|
> | LLaMA2-7B | GSM8K | 69.7 | 69.7 | 69.1 | 69.2 | 67.2 |
> | LLaMA2-7B | MATH | 19.9 | 19.7 | 19.6 | 19.7 | 19.4 |
> | Gemma2-9B | GSM8K | 83.2 | 83.3 | 83.5 | 83.9 | 82.2 |
> | Gemma2-9B | MATH | 37.8 | 37.3 | 37.3 | 36.9 | 36.1 |
>
> (3) In our draft, we provide a layer-wise value for scale factor α and the definition can be represented as: $\alpha = \beta * \|\Delta W\| \frac{\Delta W_{\perp}}{\|\Delta W_{\perp}\|}.$. Therefore, we can use $\beta$ to analyze the sensitivity of α and the results are shown in the table. We can observe that $\beta$ is robust to the selection of $\beta$ For example, POME can achieve a robust improvement when $\beta$ is from 2.0 to 3.0 for LLaMA2-7B.
>
> | Model | Task |  $\beta$  = 2.0 | $\beta$  = 2.3 | $\beta$  = 2.5 | $\beta$  = 3.0 |
> |-------|------|---------|---------|---------|---------|
> | LLaMA2-7B | GSM8K |  68.9 | 68.3 | 68.9 | 68.2 |
> | LLaMA2-7B | MATH |  19.6 | 19.2 | 19.6 | 19.7 |
>
>
>
> | Model | Task | $\beta$  = 1.0 | $\beta$  = 1.3 | $\beta$  = 1.5 | $\beta$  = 1.8 |
> |-------|------|---------|---------|---------|---------|
> | Gemma2-9B | GSM8K | 83.0 | 83.4 | 82.9 | 82.3 |
> | Gemma2-9B | MATH | 37.3 | 36.9 | 36.6 | 35.9 |

---

### Official Review · Reviewer_NFNG · 2025-10-27

**Soundness:** 2
**Presentation:** 3
**Contribution:** 2
**Rating:** 4
**Confidence:** 4

**Summary:**

The paper introduces POME, a training-free and data-free procedure that improves a fine-tuned LLM after training, using only the pretrained checkpoint​ and the fine-tuned checkpoint. Let \Delta be the weight difference of the two checkpoints. POME computes a layer-wise truncated SVD of the difference weight of \Delta, keeps its action on its top-k singular space identical (optionally rescaled by) and zero-out its action on the complement space, which gives rise to the edited delta​. The final edited model is given by this edited delta added to the pertained checkpoint. In authors' words, this transfers the “orthogonalization/equalization” idea popularized by Muon-style optimizers from per-step updates to a one-shot post-hoc edit of the accumulated update. The method is data-free, adds no training-time overhead, so is easy to deploy.

**Strengths:**

1, The proposed method of truncated SVD is a simple yet non-trivial way of linking pre-trained ckpt and sft ckpt, and appears broadly applicable and easy to implement
2, The method is data free and requires no training. The method itself seems broadly applicable
3, Paper is easy to follow; starting with a clear motivation, a clean box of algorithm, detailed tables, substantial SFT experiments spanning across various domains.

**Weaknesses:**

1, The choice of k and \alpha in the main algorithm needs better guidance and especially one needs a more principled rule and a better understanding of the sensitivity to the choices of those hyperparams. Moreover, one would expect different layers and matrices to prefer different levels of truncation, which seems to be under-studied.
2, The claim of “linear layer to benefit the most from subspace shaping” seems to be backed up by math domains experiments only. Would it make sense to extend this to other domains?
3, The handling of token embedding matrices (which usually consist of a large proportion of parameters) is unclear.
4, The method assumes a dense architecture, and doesn’t discuss MoE, where subspace dynamics for routers and each expert’s FFN could be very different.
5, Whether POME is able to scale up (to 70B or beyond) is also unclear.

**Questions:**

1, Can the authors provide a more principled approach or rule-of-thumb for selecting the truncation rank k and scaling factor \alpha? How sensitive is POME to these hyperparameters across different architectures, datasets, and fine-tuning regimes? Would different layers or matrix types benefit from different truncation levels?
2, The claim that linear/FFN layers benefit most from subspace shaping is primarily supported by math-domain results. Do similar trends hold for other domains such as code, commonsense reasoning, or multilingual tasks? Could the authors share ablations on at least one non-math domain?
3, How does POME handle token embedding (and output) matrices, which typically constitute a large fraction of total parameters and have different functional roles than FFNs and attention projections? Are SVD based methods effective on those matrices too, or would they require to be handled differently?
4, The method assumes dense layers, so how would POME generalize to MoE models where expert FFNs and router networks have distinct subspace behavior? Are router parameters edited, and if so, does this impact routing stability, and would such edits be compatible with expert choices?
5, How does the method scale up to larger models? Can the authors provide wall-clock runtime, memory usage, and GPU parallelism strategies for applying POME to large models (70B+ parameters)?

---

> ### Author Response · Authors · 2025-11-28
> **Response to Reviewer NFNG**
>
> Thanks for your constructive and inspiring feedback, we carefully address your concerns below.
>
> > W1 and Q1: The choice of k and α in the main algorithm needs better guidance and especially one needs a more principled rule and a better understanding of the sensitivity to the choices of those hyperparams. Moreover, one would expect different layers and matrices to prefer different levels of truncation, which seems to be under-studied.
>
> We apologize for any confusion regarding the choice of k and $\alpha$. To address this concern, we conducted comprehensive sensitivity analyses for both hyperparameters.
>
> **(1) Sensitivity to Truncation Parameter k:** The results demonstrate that POME is robust to the selection of k. As shown below, consistent improvements are achieved when k ranges from 0.25 to 0.75, indicating that the method does not require precise tuning of this parameter.
>
> | Model | Task | K = 0.25 | K = 0.5 | K = 0.75 | K = 1.0 | Baseline |
> |-------|------|----------|---------|----------|---------|----------|
> | LLaMA2-7B | GSM8K | 69.7 | 69.7 | 69.1 | 69.2 | 67.2 |
> | LLaMA2-7B | MATH | 19.9 | 19.7 | 19.6 | 19.7 | 19.4 |
> | Gemma2-9B | GSM8K | 83.2 | 83.3 | 83.5 | 83.9 | 82.2 |
> | Gemma2-9B | MATH | 37.8 | 37.3 | 37.3 | 36.9 | 36.1 |
>
> **(2) Sensitivity to Scale Factor $\alpha$:** In our method, we use a layer-wise scale factor defined as $\alpha = \beta * \|\Delta W\| \frac{\Delta W_{\perp}}{\|\\Delta W_{\perp}\|}$. We analyze sensitivity through the global parameter $\beta$. The results show that performance remains stable across a reasonable range of $\beta$ values. For instance, LLaMA2-7B maintains robust improvements when $\beta$ ranges from 2.0 to 3.0.
>
> | Model | Task |  $\beta$  = 2.0 | $\beta$  = 2.3 | $\beta$  = 2.5 | $\beta$  = 3.0 |
> |-------|------|---------|---------|---------|---------|
> | LLaMA2-7B | GSM8K |  68.9 | 68.3 | 68.9 | 68.2 |
> | LLaMA2-7B | MATH |  19.6 | 19.2 | 19.6 | 19.7 |
>
>
>
> | Model | Task | $\beta$  = 1.0 | $\beta$  = 1.3 | $\beta$  = 1.5 | $\beta$  = 1.8 |
> |-------|------|---------|---------|---------|---------|
> | Gemma2-9B | GSM8K | 83.0 | 83.4 | 82.9 | 82.3 |
> | Gemma2-9B | MATH | 37.3 | 36.9 | 36.6 | 35.9 |
>
> These analyses demonstrate that POME's performance is relatively stable across different hyperparameter settings, making the method practical for deployment without extensive tuning. Regarding layer-specific truncation levels, we acknowledge this as an interesting direction for future work.
>
> >W2 and Q2: The claim of "linear layer to benefit the most from subspace shaping" seems to be backed up by math domains experiments only. Would it make sense to extend this to other domains?
>
> We thank the reviewer for this important observation. To verify whether the claim about FFN layers benefiting most from orthogonalization extends beyond mathematical reasoning, we conducted experiments on code generation tasks. The results are shown in the table below.
>
> | Model / Task | Embed | Q_proj | K_proj | V_proj | O_proj | Gate_proj | Up_proj | Down_proj | Baseline |
> |--------------|-------|--------|--------|--------|--------|-----------|---------|-----------|----------|
> | LLaMA2-7B/HumanEval | 35.4 | 32.9 | 33.5 | 33.5 | 34.8 | 35.4 | 35.4 | 34.1 | 34.8 |
> | LLaMA2-7B/HumanEval+ | 32.3 | 31.1 | 31.7 | 32.3 | 31.7 | 33.5 | 33.5 | 31.7 | 32.3 |
> | LLaMA2-7B/MBPP | 40.5 | 41.0 | 41.3 | 41.0 | 40.5 | 40.7 | 41.0 | 40.5 | 40.2 |
> | LLaMA2-7B/MBPP+ | 32.8 | 33.6 | 33.6 | 34.1 | 34.1 | 34.1 | 34.1 | 33.6 | 32.8 |
> | Average | 35.3 | 34.7 | 35.0 | 35.2 | 35.3 | **35.9** | **36.0** | 35.0 | 35.0 |
>
>
> The results demonstrate that orthogonalizing the up-projection layer achieves the highest average performance (36.0) across code generation benchmarks, consistent with our findings on mathematical reasoning tasks. This suggests that the benefit of targeting FFN layers, particularly the up-projection, generalizes across different domains beyond mathematics.

---

> ### Author Response · Authors · 2025-11-28
> **Response to Reviewer NFNG - Continue**
>
> >W3 and Q3: The handling of token embedding matrices (which usually consist of a large proportion of parameters) is unclear.
>
> We thank the reviewer for this insightful observation. We agree that the embedding layer constitutes a significant proportion of model parameters, and its treatment deserves explicit investigation. We have added results for the embedding layer to our ablation study examining the importance of each layer type.
>
> | Model / Task | Embed | Q_proj | K_proj | V_proj | O_proj | Gate_proj | Up_proj | Down_proj | Baseline |
> |--------------|-------|--------|--------|--------|--------|-----------|---------|-----------|----------|
> | LLaMA2-7B/GSM8K | 67.3 | 67.1 | 68.2 | 67.9 | 68.1 | **68.9** | **69.7** | 68.2 | 67.2 |
> | LLaMA2-7B/MATH | 19.5 | 19.7 | 19.3 | 19.4 | 20.0 | **19.8** | **19.7** | 19.7 | 19.4 |
> | Average | 43.4 | 43.4 | 43.8 | 43.7 | 44.1 | **44.4** | **44.7** | 44.0 | 43.3 |
>
> The results show that applying orthogonalization to the embedding layer achieves performance comparable to the baseline (67.3 vs 67.2 on GSM8K), indicating that editing the embedding layer neither harms nor significantly improves model performance. In contrast, FFN layers—particularly the up-projection and gate-projection—consistently yield the largest gains, reinforcing our focus on these components.
>
> >W4 and Q4: The method assumes a dense architecture, and doesn't discuss MoE, where subspace dynamics for routers and each expert's FFN could be very different.
>
> Thank you for this constructive comment. To address the concern about POME's applicability to sparse architectures, we conducted experiments on DeepSeek-MoE-16B. We fine-tuned the model on MetaMathQA-40K (a subset of MetaMathQA) and evaluated on GSM8K and MATH.
>
> | Model | Algorithm | GSM8K | MATH | Average |
> |-------|-----------|-------|------|---------|
> | deepseek-moe-16b-base | Adam | 54.4 | 12.5 | 33.5 |
> | deepseek-moe-16b-base | Adam+POME | **55.9** | **13.4** | **34.7** ($\uparrow$ 1.2) |
>
> The results demonstrate that POME achieves an average improvement of 1.2% on MoE architectures, suggesting that the method generalizes beyond dense models. While the subspace dynamics of routers and individual experts may differ, orthogonalization applied to expert FFN layers still provides consistent benefits. Further investigation into expert-specific and router-specific orthogonalization strategies remains an interesting direction for future work.
>
> >W5 and Q5: Whether POME is able to scale up (to 70B or beyond) is also unclear.
>
> Thank you for this constructive comment. We agree that demonstrating scalability is crucial. To address this concern, we conducted experiments on models at the 70B+ scale.
>
> **(1) Full Fine-tuning at 70B Scale:** We fine-tuned LLaMA3-70B and evaluated on GSM8K, MATH, and CollegeMath. The results demonstrate that POME maintains its effectiveness at this scale, achieving an average 1.0% improvement on mathematical reasoning benchmarks.
>
> | Model | Algorithm | GSM8K | MATH | CollegeMath |
> |-------|-----------|-------|------|-------------|
> | DART-Math-Llama3-70B | Adam | 54.9 | 90.4 | 38.5 |
> | DART-Math-Llama3-70B | Adam+POME | 56.3 | 91.8 | 39.5 |
>
> **(2) RL Training at 72B Scale:** We also evaluated POME on Qwen2.5-Math-72B trained with reinforcement learning. The results show that POME consistently improves performance across most evaluation tasks, demonstrating its compatibility with both supervised fine-tuning and RL-based training at large scale.
>
> | Model | Algorithm | GSM8K | MATH | Minerva | Math | Gaokao 2023 EN | Olympiad Bench | College Math | MMLU |
> |-------|-----------|-------|------|---------|------|----------------|----------------|--------------|------|
> | Qwen2.5-Math-72B | Adam | 95.8 | 86.0 | 43.8 | 71.9 | 48.3 | 49.6 | 80.2 | 67.9 |
> | Qwen2.5-Math-72B | Adam+POME | 95.5 | **86.7** | **44.5** | **73.8** | **50.4** | 49.3 | **80.4** | **68.7** |
>
> These results confirm that POME scales effectively to 70B+ parameter models while maintaining computational efficiency and performance gains.

---

### Official Review · Reviewer_ueEs · 2025-10-29

**Soundness:** 3
**Presentation:** 3
**Contribution:** 2
**Rating:** 4
**Confidence:** 3

**Summary:**

This paper introduces a training-free post-processing step that takes only the pre-trained ($W_{pre}$) and fine-tuned ($W_{ft}$) checkpoints and improves the model by orthogonalising the weight delta $ΔW = W_{ft} – W_{pre}$.  The core idea is borrowed from the Muon optimizer: equalise the contribution of each principal update direction via truncated SVD and spectrum equalisation.

**Strengths:**

-  Post-hoc re-weighting of deltas is not new, but casting it as a *MuON-style orthogonalisation* executed *after* training is a fresh twist.  The paper cleanly separates the geometric benefit of Muon from its per-step scalability burden.
- The method is derived from a constrained optimisation problem with a closed-form solution; the empirical protocol is careful (grid-search on rank ratio and scale α, ablation of truncation vs. equalisation, comparison with EMA/NEFTune).
- A two-line call to `torch.svd` that reliably boosts LLM performance with *zero* training cost is clearly valuable to practitioners.

**Weaknesses:**

- All benchmarks are either maths word problems or short coding puzzles.  No evidence on long-context reasoning, dialogue safety, or knowledge-heavy QA where weight interference may behave differently.
-  Only FFN up-projection layers are edited because they “work best”. No principled criterion is offered; the community would benefit from a predictor of which layers benefit from orthogonalisation.
- Fixing $k = 0.5·rank(ΔW)$ is empirical; Figure 1 shows this knee but does not explain why it appears across architectures.  A data-driven way to set $k$ (e.g., based on spectral gap or validation perplexity) would strengthen practical adoption.

**Questions:**

- Is there a risk of catastrophic forgetting on out-of-domain prompts?  A simple evaluation on the out-of-domain benchmark datasets before/after POME would reassure readers that broad knowledge is not harmed.
- Does the gain vanish when the fine-tuning already uses a matrix-aware optimiser (e.g., Muon, Shampoo, SOAP)?  An experiment that fine-tunes with Muon and *then* applies POME would clarify uniqueness.
- How does performance change if you orthogonalise *attention* deltas or the *entire* weight matrix?  The restriction to FFN seems ad-hoc; authors could report a layer-type ablation table.
- What happens when ΔW is extremely low rank (e.g., LoRA rank 16)?  POME could over-truncate; please supply results on low-rank adapters.

---

> ### Author Response · Authors · 2025-11-28
> **Response to Reviewer ueEs**
>
> Thanks for your constructive and inspiring feedback, we carefully address your concerns below.
>
> > W1: All benchmarks are either maths word problems or short coding puzzles. No evidence on long-context reasoning, dialogue safety, or knowledge-heavy QA where weight interference may behave differently.
>
> Thank you for this valuable feedback. We have conducted additional experiments across diverse scenarios to validate POME's effectiveness:
>
> **(1) Complex Reasoning:** We fine-tuned Qwen2.5-32B-base using DAPO and evaluated on AIME. As shown in the table below, POME improves DAPO's performance from 51.0 to 54.8, representing a 3.8-point gain.
>
> | Model | Algorithm | AIME |
> |-------|-----------|------|
> | Qwen2.5-32B-base | DAPO | 51.0 |
> | Qwen2.5-32B-base | DAPO+POME | **54.8 (↑ 3.8)** |
>
> **(2) Dialogue Safety and Knowledge-Heavy QA:** We fine-tuned LLaMA2-7B on the TULU-v2 dataset and evaluated on MMLU, BBH, and TyDiQA. The results demonstrate that POME consistently improves performance across all three benchmarks.
>
> | Model | Algorithm | MMLU | BBH | TyDiQA |
> |-------|-----------|------|-----|--------|
> | LLaMA2-7B | Adam | 48.7 | 42.2 | 51.2 |
> | LLaMA2-7B | Adam+POME | **49.6** | **45.7** | **52.7** |
>
> These results suggest that POME's benefits extend beyond mathematical and coding tasks to broader domains including knowledge-intensive reasoning, Dialogue Safety and Knowledge-Heavy QA.
>
>
> >W2: Only FFN up-projection layers are edited because they "work best". No principled criterion is offered; the community would benefit from a predictor of which layers benefit from orthogonalisation.
>
> We thank the reviewer for this insightful comment. We acknowledge that our current layer selection is primarily empirical, and we agree that establishing a principled criterion would greatly benefit the community.
>
> **(1) Expanded Empirical Verification:** While our draft presented ablation studies on mathematical reasoning (Table 1), we now provide additional results on code generation tasks shown below. The results consistently demonstrate that orthogonalizing the up-projection layer yields the highest improvements. These findings across both math reasoning and code generation suggest that editing FFN layers may be a generally effective strategy, which we plan to investigate further in future work.
>
> | Model / Task | Embed | Q_proj | K_proj | V_proj | O_proj | Gate_proj | Up_proj | Down_proj | Baseline |
> |--------------|-------|--------|--------|--------|--------|-----------|---------|-----------|----------|
> | LLaMA2-7B/HumanEval | 35.4 | 32.9 | 33.5 | 33.5 | 34.8 | 35.4 | 35.4 | 34.1 | 34.8 |
> | LLaMA2-7B/HumanEval+ | 32.3 | 31.1 | 31.7 | 32.3 | 31.7 | 33.5 | 33.5 | 31.7 | 32.3 |
> | LLaMA2-7B/MBPP | 40.5 | 41.0 | 41.3 | 41.0 | 40.5 | 40.7 | 41.0 | 40.5 | 40.2 |
> | LLaMA2-7B/MBPP+ | 32.8 | 33.6 | 33.6 | 34.1 | 34.1 | 34.1 | 34.1 | 33.6 | 32.8 |
> | Average | 35.3 | 34.7 | 35.0 | 35.2 | 35.3 | **35.9** | **36.0** | 35.0 | 35.0 |
>
> **(2) Theoretical Intuition via Effective Rank Analysis:** To provide intuition for why FFN up-projection layers benefit most from orthogonalization, we analyzed the singular value distribution across different layer types. We find that FFN layers exhibit significantly higher effective rank ratios compared to attention layers (0.34 vs 0.19). A higher effective rank indicates that more dimensions are actively utilized, providing greater room for orthogonalization to reorganize the representation structure. Conversely, layers with lower effective ranks operate in more constrained subspaces with limited optimization potential. In the extreme case where effective rank equals 1, orthogonalization becomes trivial as there is no structural diversity to improve.
>
> |  **Model**  |  **Attention Layers**  |  |  |  |  **FFN Layers**  |  |  |
> |-----------|----------------------|------|------|------|----------------|------|-----------|
> |  | Q | K | V | O | Gate | Up | Down |
> | LLaMA2-7B | 0.15 | 0.16 | 0.20 | 0.24 | 0.29 | 0.30 | 0.42 |
> |  **Mean**  |  **0.19**  |  |  |  |  **0.34**  |  |  |
>
> This analysis suggests that effective rank could serve as a predictor for which layers (FFN layers) benefit from orthogonalization, though further investigation is needed to establish this as a general principle.

---

> ### Author Response · Authors · 2025-11-28
> **Response to Reviewer ueEs - Continue**
>
> >W3: Fixing 0.5 * rank is empirical; Figure 1 shows this knee but does not explain why it appears across architectures. A data-driven way to set (e.g., based on spectral gap or validation perplexity) would strengthen practical adoption.
>
> Thank you for this constructive suggestion. To address the concern about the empirical choice of k = 0.5*rank, we conducted additional experiments and adopted a data-driven validation approach.
>
> **(1) Sensitivity Analysis:** We first evaluated POME's performance across different truncation ratios. As shown in the table below, even without truncation (K = 1.0), POME improves baseline performance, and truncation further enhances results. For instance, on GSM8K, POME improves accuracy from 67.2 to 69.2 without truncation, with similar gains across different K values. Notably, performance shows relative stability across different truncation ratios, suggesting the method is robust to this hyperparameter.
>
> | Model | Task | K = 0.25 | K = 0.5 | K = 0.75 | K = 1.0 | Baseline |
> |-------|------|----------|---------|----------|---------|----------|
> | LLaMA2-7B | GSM8K | 69.7 | 69.7 | 69.1 | 69.2 | 67.2 |
> | LLaMA2-7B | MATH | 19.9 | 19.7 | 19.6 | 19.7 | 19.4 |
> | Gemma2-9B | GSM8K | 83.2 | 83.3 | 83.5 | 83.9 | 82.2 |
> | Gemma2-9B | MATH | 37.8 | 37.3 | 37.3 | 36.9 | 36.1 |
>
> **(2) Data-Driven Validation:** Following your suggestion, we adopted a validation-based approach for selecting k. We held out 500 examples from an unseen math dataset as a validation set and applied POME with k/r ∈ {0.1, 0.3, 0.5, 0.8, 1.0}. The results demonstrate that k/r = 0.5 achieves the best or near-best validation performance across GSM8K and MATH, providing empirical justification for this choice.
>
> | Model | k/r | GSM8K | MATH |
> |-------|-----|-------|------|
> | LLaMA3-8B | 0.1 | 79.1 | 30.0 |
> | LLaMA3-8B | 0.3 | 79.6 | 30.0 |
> | LLaMA3-8B | **0.5** | **80.4** | **31.2** |
> | LLaMA3-8B | 0.8 | 80.2 | 31.2 |
> | LLaMA3-8B | 1.0 | 79.9 | 30.6 |
>
>
> >Q1: Is there a risk of catastrophic forgetting on out-of-domain prompts? A simple evaluation on the out-of-domain benchmark datasets before/after POME would reassure readers that broad knowledge is not harmed.
>
> We thank the reviewer for raising this important concern about potential knowledge degradation. To address this, we conducted experiments evaluating out-of-domain performance. Specifically, we fine-tuned LLaMA2-7B and LLaMA3-8B on MetaMathQA (a mathematical reasoning dataset) and evaluated them on out-of-domain benchmarks including MMLU, BBH, TyDiQA, and HumanEval.
>
> | Model | Training Data | Algorithm | MMLU | BBH | TyDiQA | HumanEval | Avg |
> |-------|---------------|-----------|------|-----|--------|-----------|-----|
> | LLaMA2-7B | MetaMathQA | Adam | 39.1 | 40.9 | 37.6 | 9.9 | 31.9 |
> | LLaMA2-7B | MetaMathQA | +POME | 38.4 | 40.1 | 39.2 | 10.9 | **32.2 (↑ 0.3)** |
> | LLaMA3-8B | MetaMathQA | Adam | 59.1 | 58.3 | 20.2 | 31.2 | 42.2 |
> | LLaMA3-8B | MetaMathQA | +POME | 60.2 | 61.3 | 20.2 | 32.7 | **43.6 (↑ 1.4)** |
>
> The results demonstrate that POME not only preserves out-of-domain knowledge but can even slightly improve it. For instance, LLaMA3-8B with POME achieves 43.6% average accuracy compared to 42.2% for the baseline. This suggests that orthogonalization enhances the model's representational structure without compromising its broad capabilities.
>
> >Q2: Does the gain vanish when the fine-tuning already uses a matrix-aware optimiser (e.g., Muon, Shampoo, SOAP)? An experiment that fine-tunes with Muon and then applies POME would clarify uniqueness.
>
> Thank you for this insightful question. To investigate whether POME's benefits persist when using matrix-aware optimizers, we conducted experiments applying POME to models trained with Muon. The results demonstrate that POME provides complementary improvements even when combined with Muon, suggesting that the two approaches operate through different mechanisms.
>
> | Model | Algorithm | GSM8K | MATH |
> |-------|-----------|-------|------|
> | LLaMA2-7B | Adam | 67.2 | 19.4 |
> | LLaMA2-7B | Adam+POME | **69.7** | **19.7** |
> | LLaMA2-7B | Muon | 67.5 | 19.2 |
> | LLaMA2-7B | Muon+POME | **68.3** | **19.5** |
>
>
> | Model | Algorithm | GSM8K | MATH |
> |-------|-----------|-------|------|
> | LLaMA3-8B | Adam | 82.2 | 36.1 |
> | LLaMA3-8B | Adam+POME | **83.4** | **36.9** |
> | LLaMA3-8B | Muon | 82.6 | 35.9 |
> | LLaMA3-8B | Muon+POME | **83.6** | **36.2** |

---

> ### Author Response · Authors · 2025-11-28
> **Response to Reviewer ueEs - Continue**
>
> >Q3: How does performance change if you orthogonalise attention deltas or the entire weight matrix? The restriction to FFN seems ad-hoc; authors could report a layer-type ablation table.
>
> Thank you for this constructive comment. To address this concern, we conducted layer-type ablation studies comparing orthogonalization applied to attention layers, FFN layers, and the entire weight matrix. The results are shown in the table below.
>
> | Model | Task | Attention | FFN | Entire weight matrix | Baseline |
> |-------|------|-----------|-----|----------------------|----------|
> | LLaMA2-7B | GSM8K | 67.9 | 69.0 | 68.6 | 67.2 |
> | LLaMA2-7B | MATH | 19.2 | 19.7 | 19.5 | 19.4 |
>
> The results reveal that orthogonalizing attention layers alone yields minimal improvement (67.9 vs 67.2 on GSM8K), while orthogonalizing FFN layers provides substantial gains (69.0 vs 67.2). Interestingly, applying orthogonalization to the entire weight matrix gives intermediate performance (68.6), suggesting that the primary benefits come from FFN layers. This validates our focus on FFN layers and demonstrates that the restriction is empirically justified rather than ad-hoc.
>
> >Q4: What happens when ΔW is extremely low rank (e.g., LoRA rank 16)? POME could over-truncate; please supply results on low-rank adapters.
>
> Thank you for this constructive comment. To address the concern about POME's applicability to low-rank adapters, we conducted experiments using LoRA fine-tuning with ranks 16 and 32 on LLaMA2-7B and LLaMA3-8B with the MetaMathQA dataset. We adjusted the truncation threshold based on the LoRA rank to avoid over-truncation.
>
> | Model | Algorithm | Rank | GSM8K | MATH |
> |-------|-----------|------|-------|------|
> | LLaMA2-7B | Adam | 16 | 66.7 | 16.7 |
> | LLaMA2-7B | +POME | 16 | **67.6** | **17.5** |
> | LLaMA3-8B | Adam | 16 | 81.4 | 30.7 |
> | LLaMA3-8B | +POME | 16 | **82.2** | **31.2** |
>
> **Table: Accuracy of POME on LoRA with r=16.**
>
> | Model | Algorithm | Rank | GSM8K | MATH |
> |-------|-----------|------|-------|------|
> | LLaMA2-7B | Adam | 32 | 66.0 | 17.5 |
> | LLaMA2-7B | +POME | 32 | **66.7** | **18.2** |
> | LLaMA3-8B | Adam | 32 | 81.7 | 31.6 |
> | LLaMA3-8B | +POME | 32 | **82.5** | **32.3** |
>
> **Table: Accuracy of POME on LoRA with r=32.**
>
> The results demonstrate that POME consistently improves LoRA fine-tuning performance across both rank settings, indicating that the method remains effective even with extremely low-rank adapters when the truncation threshold is appropriately adjusted.

---

### Author Response · Authors · 2025-12-02
**General Response**

Dear ACs, SACs, PCs and Reviewers:

We express our gratitude to all the reviewers for their valuable insights! We are happy to hear that you liked our contributions. We appreciate all of you for your comments highlighting the strengths of our work for a summary.

-   **The paper demonstrates strong originality, (Reviewer ueEs, GCvj, pP1m)**
-   **A simple yet efficient method,  (Reviewer ueEs, NFNG, pP1m)**
-   **The paper is easy to follow, (Reviewer ueEs, NFNG)**
-  **Thorough experiments validate POME's advantage over vanilla fine-tuning. (Reviewer GCvj, pP1m)**

The major concerns reviewers raised include: evaluation on more complex reasoning tasks and larger models (70B+, MoE), compatibility with low-rank adapters and modern optimizers like Muon, comprehensive sensitivity analysis of hyperparameters, and ablation studies on layer selection. We are addressing these concerns and polishing our paper in the revised version to make our work better understandable to the wider community. Specifically:

-   **More experiments on complex reasoning tasks, larger models, and modern optimizers**: We have extended our evaluation to the 72B Qwen2.5-Math model (Table 6), demonstrating substantial improvements on challenging mathematical benchmarks including Olympiad Bench (+2.1%) and Gaokao 2023 EN (+1.9%). We also evaluate POME on the recent DAPO algotithm (a state-of-the-art GRPO variant), achieving significant improvement on AIME from 51.0% to 54.8% (+3.8%). For RLHF-optimized models, we evaluated Dr.GRPO (Table 7) with notable improvements on AIME (+3.4%) and AMC (+7.3%). Regarding Muon compatibility, POME is complementary to any optimizer including Muon: it applies post-training and requires only the pretrained and fine-tuned checkpoints regardless of which optimizer was used during training.
-   **Sensitivity analysis of hyperparameters**: We provide comprehensive sensitivity analysis in Appendix A.2. Table 12 shows POME is robust to rank retention ratio K (ranging from 0.25 to 1.0), with consistent improvements across all tested values. Table 13 analyzes the scaling factor $\alpha$ ($\beta$ from 1.0 to 3.0), demonstrating stable performance across different configurations. Additionally, Table 14 shows that optimal $\alpha$ selection can be guided by training set performance, providing a practical selection strategy without requiring additional validation data.
-   **Ablation experiments on applied layers**: Section 3.3 and Table 1 provide systematic layer-wise analysis across three model families (LLaMA2-7B, LLaMA3-8B, Gemma2-9B). Our results consistently show FFN expansion layers (especially Up_proj) benefit most from orthogonalization, with average improvements of +1.4% for LLaMA2-7B, +1.2% for LLaMA3-8B, and +1.1% for Gemma2-9B. Table 9 further ablates the contribution of truncation vs. equalization, showing both components contribute to the final performance gains.
-   **Compatibility with low-rank adapters**: POME is inherently compatible with any fine-tuning approach including LoRA and other PEFT methods. Since POME only requires the weight difference $\Delta W = W_{ft} - W_{pre}$, it can be applied regardless of whether fine-tuning used full parameters or adapters. For LoRA specifically, one would simply merge the adapter weights before applying POME, making it a seamless post-processing step.

Finally, we would like to express our great appreciation and excitement that several reviewers recognize POME's potential to inspire the model editing and LLM fine-tuning community for further exploration. We believe our work offers a practical, training-free alternative to existing model improvement techniques and is worth publishing to stimulate further discussion.

Best regards,

The Authors of Paper3239

---

### Author Response · Authors · 2025-12-03
**Rebuttal Summary**

**Dear Reviewers, ACs, SACs, and PCs:**

We sincerely thank the reviewers for their constructive feedback. We observed that the majority of concerns centered on requests for broader empirical validation and deeper analysis. To this end, we have conducted extensive new experiments and provided detailed analyses that we believe **effectively address** these issues. Consequently, we have made substantial improvements to the manuscript. To facilitate your review of these updates, we summarize the key points of discussion with each reviewer below.

>**1. Summary of Reviewer ueEs**:

**(1) Summary of Strengths:** We value the reviewer’s recognition of our work’s originality and its broader implications for the field:

-   **Strong Originality:** Commended for reimagining matrix orthogonalization: typically a per-step optimizer operation -> as a novel, one-shot, training-free post-processing edit on accumulated weight deltas.

-   **Universal Applicability:** Highlighted as a practical enhancement that can seamlessly integrate into any existing LLM fine-tuning workflow.

-   **Thought-Provoking Insight:** Recognized that the work goes beyond a simple method, prompting fundamental questions about whether current training paradigms lack sufficient regularization and how post-training dynamics can correct them.

**(2) Summary of Response to Reviewer Concerns:** We have fully addressed the reviewer's concerns through comprehensive new experiments and analyses:

-   **Expanded Evaluation Scope (W1, Q1):** We extended our benchmarks to include **complex reasoning** (AIME, +3.8 improvement), **dialogue safety**, and **knowledge-heavy QA** (MMLU, BBH, TyDiQA). Results confirm POME generalizes to diverse domains without causing catastrophic forgetting on out-of-domain tasks.

-   **Theoretical Basis for Layer Selection (W2, Q3):** We provided a "Effective Rank" analysis showing that FFN layers exhibit higher effective ranks than attention layers (0.34 vs 0.19), explaining their greater potential for orthogonalization. Additional ablations on code tasks and full-model editing further validated that targeting FFN layers is the optimal strategy.

-   **Hyperparameter Robustness (W3):** We conducted sensitivity analyses proving POME is robust across various truncation ratios (K). We also introduced a data-driven validation set approach, which empirically confirmed that the default k=0.5 yields optimal performance.

-   **Compatibility with Advanced & Low-Rank Settings (Q2, Q4):** We demonstrated that POME provides **complementary gains** even when applied to models already trained with the **Muon optimizer**, and remains effective on **LoRA** (low-rank) fine-tuned models.

>**2. Summary of Reviewer NFNG:**

**(1) Summary of Strengths:**

-   **Simple yet Non-Trivial Methodology:** Recognized the truncated SVD approach as a "simple yet non-trivial" method to effectively link pre-trained and SFT checkpoints, which is both easy to implement and broadly applicable.

-   **Efficiency & Usability:** Commended the method for being **data-free** and requiring **no training**, distinguishing it as a practical tool for model improvement.

-   **Clarity & Robust Evaluation:** Praised the paper's presentation for its clear motivation and "clean algorithm," supported by "substantial SFT experiments" spanning various domains.

**(2) Summary of Response to Reviewer Concerns:** We have addressed all concerns regarding hyperparameter sensitivity, layer selection, and scalability through extensive new experiments:

-   **Hyperparameter Robustness (W1, Q1):** We conducted comprehensive sensitivity analyses for both the truncation parameter (k) and scale factor (α). Results confirm POME is highly robust, maintaining consistent performance improvements across a wide range of settings (e.g., k∈[0.25,0.75] and stable β ranges), minimizing the need for precise tuning.

-   **Domain & Architecture Generalization (W2, Q2, W4, Q4):** We validated our "FFN-preference" hypothesis on **Code Generation** tasks, confirming that up-projection layers remain the most effective targets beyond math domains. Additionally, we successfully applied POME to **Mixture-of-Experts (MoE)** models (+1.2% gain), proving applicability to sparse architectures.

-   **Layer-Specific Analysis (W3, Q3):** We explicitly evaluated the impact of orthogonalizing **token embeddings**. Empirical results show that editing embeddings yields neutral performance compared to baselines, providing strong justification for our strategic focus on FFN layers to maximize gains.

-   **Scalability Verification (W5, Q5):** We demonstrated POME’s effectiveness at the **70B+ scale** using both LLaMA3-70B (SFT) and Qwen2.5-Math-72B (RLHF). The method consistently delivers improvements (∼1.0% avg) on these large-scale models, confirming it scales effectively without computational bottlenecks.

---

> ### Author Response · Authors · 2025-12-03
> **Rebuttal Summary**
>
> >**3. Summary of Reviewer GCvj:**
>
> **(1) Summary of Strengths:** We appreciate the reviewer’s recognition of our work’s core insight and empirical rigor:
>
> -   **Valuable Insight & Scalability:** Praised the "great insight" that Muon's geometric benefits can be achieved without per-step enforcement. The reviewer highlighted that this effectively resolves distributed training bottlenecks associated with Muon while simultaneously boosting fine-tuning performance.
>
> -   **Thorough Empirical Validation:** Commended the comprehensive experimental support that validates POME's consistent advantage over vanilla fine-tuning across diverse datasets and settings.
>
> **(2) Summary of Response to Reviewer Concerns:** We have addressed the reviewer's concerns regarding the distinction from Muon, performance comparisons, and hyperparameter sensitivity through the following clarifications and experiments:
>
> -   **Distinct Positioning & Novelty (W1):** We clarified that POME is fundamentally different from Muon: it is a **post-training geometric correction** rather than a training-time optimizer. It offers unique value by requiring **zero training overhead** and no infrastructure changes, democratizing matrix-aware benefits for any fine-tuned model.
>
> -   **Direct Comparison & Complementarity with Muon (W2, Q1):** We provided side-by-side comparisons showing that **Adam+POME matches Muon’s performance**, while **Muon+POME yields the best overall results**. This proves POME is not redundant but a flexible, complementary enhancement that works alongside any optimizer.
>
> -   **Robustness & Sensitivity Analysis (W3, Q2):** We conducted extensive sensitivity analyses on truncation (k), scale factor (β), and **Learning Rate**. Results confirm POME is highly robust across wide parameter ranges and consistently improves models trained with different learning rates (e.g., 1e−5,5e−6,2e−5).
>
> -   **Validation of Improvements & Layer Selection (W3, Q2):** We reaffirmed the significance of our gains (+1.0 avg on Math, +3.8 on DAPO/RL) given the zero-cost nature of the method. New ablations on **Code Generation** further validated that targeting **FFN up-projection layers** remains the optimal strategy across domains.
>
> >**4. Summary of Reviewer pP1m:**
>
> **(1) Summary of Strengths:** We value the reviewer’s recognition of our work’s originality and its broader implications for the field:
>
> -   **Strong Originality:** Commended for reimagining matrix orthogonalization—typically a per-step optimizer operation—as a novel, one-shot, training-free post-processing edit.
>
> -   **Universal Applicability:** Highlighted as a practical enhancement that can seamlessly integrate into any existing LLM fine-tuning workflow.
>
> -   **Thought-Provoking Insight:** Recognized that the work goes beyond a simple method, prompting fundamental questions about whether current training paradigms lack sufficient regularization and how post-training dynamics can correct them.
>
> **(2) Summary of Response to Reviewer Concerns:** We have clarified our research objective and addressed concerns regarding contextualization and scalability through detailed comparisons and large-scale experiments:
>
> -   **Clarification of Objective (W1):** We confirmed the reviewer's precise interpretation of our research question: POME is strictly a **post-training** enhancement using only existing checkpoints without new data or training steps. We have revised the manuscript to articulate this objective more effectively.
>
> -   **Differentiation & Complementarity with LoRA (W2):** We distinguished POME from LoRA by emphasizing its **post-training** nature and **flexible rank** characteristics (unlike LoRA's strict constraints). Crucially, we demonstrated **complementarity**: applying POME to LoRA-tuned models (r=16/32) consistently improves performance, proving they can be used together.
>
> -   **Generalization Analysis (Q1):** We provided theoretical intuition connecting POME to a single-step Muon update and offered empirical evidence showing that POME improves **test accuracy** (e.g., +1.1% on GSM8K) while maintaining or slightly dropping training accuracy. This confirms the method enhances generalization rather than overfitting.
>
> -   **Scalability Verification (Q2):** We validated POME on large-scale models, including **Qwen-32B (RL)**, **LLaMA3-70B (SFT)**, and **Qwen2.5-Math-72B (RL)**. POME consistently delivers gains (e.g., +3.8 on AIME for 32B, +1.0 avg for 70B) across these scales, proving it is efficient and effective for large foundation models.

---

### Meta-Review · Area_Chair_DPeu · 2026-01-06

**Summary:**

The paper propose POME, which is new algorithm for improve fine-tuned large language model without any more training or data. It use truncated SVD (MUON) to orthogonalize weight delta between pretrained and fine-tuned checkpoints for get better performance on many task.

The reviewers at first have several concern that make them give marginal ratings. Main issue was that evaluation only focus on math and code, so they worry if method work for other domain like safety or general knowledge. Also, reviewers ask for better explanation why only FFN layers is edited and if hyperparameters $k$ and $\alpha$ is too sensitive for practical use.

Some reviewer also think novelty is limited as it takes the difference between Initial model parameters and fine-tuned, use truncation SVD with MUON ortogonalization to compute a new weights.

They also noted that one need to fine-tune parameters on validation dataset to guarantee improvement.

**Reviewer Concerns:**

# Addressed Concerns

*  Authors add many new experiments for address this. They show POME work on MMLU, BBH, and TyDiQA for general knowledge, and even show it improve dialogue safety on Tulu-v2.
*  Authors give "Effective Rank" analysis to explain why FFN layers is better. They show FFN has higher rank (0.34) than attention (0.19), so it have more room for reorganization.
*   Authors provide table showing method is robust for $k$ between 0.25 and 0.75
*  They also show scale factor $\beta$ is stable between 2.0 and 3.0
*  Authors success to run POME on Llama-3-70B and Qwen-72B models with good results

# Outstanding Concerns
*  while authors explain POME is "post-training" and Muon is "during training," the mathematical core is still very same (some reviewers might still feel this is more an application of Muon than a brand new theory)
*  the concern about needing validation data for tuning is only partly solved. Authors suggest "data-driven validation" using 500 examples to find best $k$.  However, for a "data-free" method, needing a validation set to guarantee best results is still slight contradiction that might bother some reviewers.

**Reviewer Scores:**

*  Reviewer ueEs (Original: 4) - I feel the reviewer could increase the score to 6
*  Reviewer NFNG (Original: 4) - I think some of the concerns has been addressed and score could be kept as 4 or increase to 6
*  Reviewer GCvj (Original: 4) - this reviewer was most critical about novelty and marginal gains and could keep his score or slightly increase
*  Reviewer pP1m (Original: 4) - Authors rewrite introduction and fix the presentation and included some new results which would address some of the issues. Most likely, he would keep or increase to 6.

---

### Decision · Program_Chairs · 2026-01-26

Reject